# Cardiac sympathetic neurons are additional cells affected in genetically determined arrhythmogenic cardiomyopathy

Induja Perumal Vanaja[1,2], Arianna Scalco[2,3], Marco Ronfini[2,3], Anna Di Bona[1,2], Camilla Olianti[4], Stefania Rizzo[1], Stephen P. Chelko[5,6], Domenico Corrado[1], Leonardo Sacconi[4,7], Cristina Basso[1], Marco Mongillo[2,3] and Tania Zaglia[2,3]

[1]*Department of Cardiac, Thoracic, Vascular Sciences and Public Health, University of Padova, Padova, Italy*

[2]*Veneto Institute of Molecular Medicine (VIMM), Padova, Italy*

[3]*Department of Biomedical Sciences, University of Padova, Padova, Italy*

[4]*Institute of Clinical Physiology (IFC), National Research Council, Florence, Florence, Italy*

[5]*Department of Medicine, Johns Hopkins University, School of Medicine, Baltimore, MD, USA*

[6]*Department of Biomedical Sciences, Florida State University, College of Medicine, Tallahassee, FL, USA*

[7]*Institute for Experimental Cardiovascular Medicine, University Heart Center and Medical Faculty, University of Freiburg, Freiburg, Germany*

Handling Editors: Harold Schultz & David Paterson

The peer review history is available in the Supporting Information section of this article (https://doi.org/10.1113/JP286845#support-information-section).

**Abstract**  Arrhythmogenic cardiomyopathy (AC) is a familial cardiac disease, mainly caused by mutations in desmosomal genes, which accounts for most cases of stress-related arrhythmic sudden death, in young and athletes. AC hearts display fibro-fatty lesions that generate the arrhythmic substrate and cause contractile dysfunction. A correlation between physical/emotional stresses and arrhythmias supports the involvement of sympathetic neurons (SNs) in the disease, but this has not been confirmed previously. Here, we combined molecular, *in vitro* and *ex vivo* analyses to determine the role of AC-linked DSG2 downregulation on SN biology and assess cardiac sympathetic innervation in desmoglein-2 mutant (*Dsg2^{mut/mut}*) mice. Molecular assays showed that SNs express DSG2, implying that DSG2-mutation carriers would harbour the mutant protein in SNs. Confocal

---

I. P. Vanaja and A. Scalco contributed equally to this work.

immunofluorescence of heart sections and 3-D reconstruction of SN network in clarified heart blocks revealed significant changes in the physiologialc SN topology, with massive hyperinnervation of the intact subepicardial layers and heterogeneous distribution of neurons in fibrotic areas. Cardiac SNs isolated from *Dsg2^{mut/mut}* neonatal mice, prior to the establishment of cardiac innervation, show alterations in axonal sprouting, process development and distribution of varicosities. Consistently, virus-assisted DSG2 downregulation replicated, in PC12-derived SNs, the phenotypic alterations displayed by *Dsg2^{mut/mut}* primary neurons, corroborating that AC-linked *Dsg2* variants may affect SNs. Our results reveal that altered sympathetic innervation is an unrecognized feature of AC hearts, which may result from the combination of cell-autonomous and context-dependent factors implicated in myocardial remodelling. Our results favour the concept that AC is a disease of multiple cell types also hitting cardiac SNs.

(Received 30 April 2024; accepted after revision 5 July 2024; first published online 1 August 2024)

**Corresponding author** T. Zaglia: Department of Biomedical Sciences, University of Padova, Via Ugo Bassi 58/B, 35122, Padova, Italy and Veneto Institute of Molecular Medicine, via Orus 2, 35129, Padova, Italy.    Email: tania.zaglia@unipd.it

**Abstract figure legend** Cardiac sympathetic neurons express desmoglein-2 (DSG2) and harbour mutations in DSG2-linked arrhythmogenic cardiomyopathy. DSG2 mutations affect sympathetic neuron biology (i.e. reduced axonal sprouting, irregular distribution of varicosities) and result in aberrant cardiac innervation. Hyperinnervation of DSG2 mutant hearts appears before structural myocardial remodelling and worsens along with disease progression.

## Key points

- Arrhythmogenic cardiomyopathy is a genetically determined cardiac disease, which accounts for most cases of stress-related arrhythmic sudden death.
- Arrhythmogenic cardiomyopathy linked to mutations in desmoglein-2 (DSG2) is frequent and leads to a left-dominant form of the disease.
- Arrhythmogenic cardiomyopathy has been approached thus far as a disease of cardiomyocytes, but we here unveil that DSG2 is expressed, in addition to cardiomyocytes, by cardiac and extracardiac sympathetic neurons, although not organized into desmosomes.
- AC-linked DSG2 downregulation primarily affect sympathetic neurons, resulting in the significant increase in cardiac innervation density, accompanied by alterations in sympathetic neuron distribution.
- Our data supports the notion that AC develops with the contribution of several 'desmosomal protein-carrying' cell types and systems.

## Introduction

Arrhythmogenic cardiomyopathy (AC) is a genetic cardiac disease at high risk of stress-related arrhythmias accounting for most cases of sudden death, predominantly in young and athletes (Corrado et al., 2001; Corrado et al., 2006; Mazzanti et al., 2016; Pilichou et al., 2016;

**Induja Perumal Vanaja** is pursuing her PhD in Translational Specialistic Medicine 'G.B. Morgagni' at the University of Padova, funded by Marie Skłodowska-Curie Actions Cofund. Her Biomedical Engineering background and laboratory experience have provided her with interdisciplinary skills allowing her to investigate how arrhythmogenic cardiomyopathy-linked mutations affect cardiac sympathetic neurons using *ex vivo* and *in vitro* experiments and advanced imaging techniques. Her future goal is to translate her research to clinics for diagnostics and prevention purposes. **Arianna Scalco**, earned her master's in medical biotechnology and PhD in Cardiovascular Sciences at the University of Padova. She studied stromal cells and cardiac sympathetic neurons in AC using *ex vivo* and *in vitro* approaches. At Johns Hopkins University, she investigated US Food and Drug Administration-approved drugs for AC treatment. Her passion for autonomic dysfunction in cardiovascular diseases led her to Oregon Health & Science University, aiming to translate her research from bench to bedside.

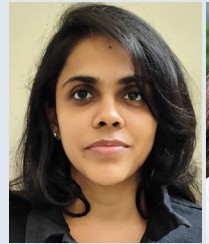
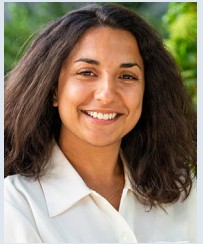

Priori et al., 2015; Thiene et al., 1988). Over 50% of inherited AC cases are the result of pathogenic variants in genes that encode desmosomal proteins: desmoglein-2 (DSG2), desmoplakin (DSP), junctional plakoglobin (JUP), plakophilin-2 (PKP2) and desmocollin-2 (DSC2). The cardiac desmosome is a mechanical and electrical junctional complex, connecting adjacent cardiomyocytes (Gerull et al., 2004; Kim et al., 2019; McKoy et al., 2000; Pilichou et al., 2006; Syrris et al., 2006). The pathological hallmarks of AC hearts are cardiomyocyte death, tissue inflammation and myocardial replacement with focal fatty lesions and diffuse fibrosis (Asimaki & Saffitz, 2014; Asimaki et al., 2015; Basso et al., 2009; Basso et al., 2011; Calore et al., 2015; de Coster et al., 2018; Kant et al., 2015; Moncayo-Arlandi et al., 2016; Patel & Green, 2014; Pilichou et al., 2009; Rasmussen et al., 2014). Such fibro-fatty scars compromise heart contractility and generate the substrate favouring arrhythmias, which are usually triggered by physical or emotional stresses. These latter conditions are associated with increased activity of the sympathetic nervous system and are recognized as risk factors in AC progression (Agrimi et al., 2020; Corrado et al., 2001). Despite years of research, AC pathogenesis is still incompletely understood and, as such, the disease is orphan of mechanism-driven therapies. Current clinical management of AC includes exercise restriction and $\beta$-blockers/class-III anti-arrhythmics to reduce the incidence of life-threatening arrhythmias, as well as implantable cardioverter-defibrillator devices to abort fatal arrhythmias in highly arrhythmogenic patients (Bhonsale et al., 2011; Boriani et al., 2007; Corrado et al., 2010; Hodgkinson et al., 2005; Link et al., 1997; Schuler et al., 2012; Wichter et al., 2004).

The typical myocardial remodelling and the association of the disease with desmosomal gene mutations has steered AC research towards cardiomyocytes, the quintessential desmosome-carrying cells in the heart. However, several non-cardiomyocyte cell types have recently been implicated in the typical AC remodelling, including: resident cardiac stem cells (Lombardi et al., 2011); progenitor cells from the second heart field (Lombardi et al., 2009); epicardium-derived progenitors (Kohela et al., 2021; Matthes et al., 2011) and mesenchymal stromal cells (Sommariva et al., 2016; Stadiotti et al., 2021). Notably, the current literature and our data indicate that almost all cardiac and extracardiac cell types express desmosomal proteins and harbour therefore AC-linked mutant variants (Scalco et al., 2021; Sommariva et al., 2016; Stadiotti et al., 2021). On these bases, we inquired whether desmosome-linked AC may affect the entire myocardial cell network.

Based on the strict correlation between arrhythmias and physical/emotional stressors (Habecker et al., 2025; Herring et al., 2019), we tested the hypothesis that AC-linked mutations also affect cardiac sympathetic neurons (cSNs). To this aim, we inspected the state of sympathetic innervation in hearts from homozygous desmoglein-2 mutant mice ($Dsg2^{mut/mut}$ mice; Chelko et al., 2016), with germline incorporation of this AC-linked variant. Sympathetic neuron (SN) morphology and topology were analysed, at different stages of disease manifestation, by confocal immunofluorescence microscopy and multiphoton imaging of clarified heart blocks (Tolstik et al., 2024). *In vitro* analyses assessed the role of DSG2 downregulation in SNs, and inspection of human samples was used to corroborate results obtained in pre-clinical models. Our data reveal that abnormal cardiac innervation is a so far unknown feature of AC hearts, thus challenging the conception of AC as a 'disease of the cardiac desmosome' and supporting the view of AC as a 'multicellular disease'.

## Methods

### Ethical approval

All of the investigators involved in the present study understand the ethical principles under which the journal operates, and the work conducted complies with the animal ethics checklist of *The Journal of Physiology* (Grundy, 2015).

### Human tissue sample processing and immunofluorescence

We analysed postmortem stellate ganglia samples from two male patients (aged $50 \pm 3$ years) who died as a result of extracardiac causes (accidents) and did not have prior history of heart disease. Samples were archived in the historical collection of the Institute of Pathological Anatomy of the University of Padova and were acquired during routine postmortem investigations. Samples were anonymized to the investigators and were used in accordance with the directives of the national committee of Bioethics and 'Raccomandazione (2006) della Commissione dei Ministri degli Stati Membri sull'utilizzo di campioni biologici di origine umana per scopi di ricerca'. Samples were analysed by confocal immunofluorescence using the protocol previously described by Zaglia et al. (2016). The primary and secondary antibodies used in this study are listed in Tables 1 and 2.

### Experimental procedures in murine models

All procedures here described were approved by Ministry of Health (Ufficio VI), in compliance with the Animal Welfare Legislation (protocols A06E0.N.ERD and A06E0.18). All procedures were performed by certified

**Table 1. Primary antibodies used in the present study**

| Antibody | Supplier | Host | RRID | Dilution | Method |
|---|---|---|---|---|---|
| SNAP25 | Biolegend (San Diego, CA, USA) | Mouse | AB_2 566 521 | 1:1000 | IF |
| Tyrosine hydroxylase | Merck Millipore (Burlington, MA, USA) | Rabbit | AB_390 204 | 1:400/1:2500 | IF/WB |
| Desmoglein-2 (DSG2) | Abcam (Cambridge, UK) | Rabbit | Cat.No_AB150372 | 1:1000 | WB |
| Desmoglein-2 (DSG2) | Abcam | Rabbit | AB_1 924 917 | 1:50 | IF |
| Gamma Catenin [plakoglobin (JUP)] | Abcam | Rabbit | AB_2 127 989 | 1:5000 | WB |
| Plakophilin (PKP2) | Progen (Heidelberg, Germany) | Mouse | AB_2 920 694 | 1:1000 | WB |
| Desmoplakin | Progen | Mouse | AB_2 920 666 | 1:200 | WB |

**Table 2. Secondary antibodies used in the present study**

| Antibody | Supplier | Host | RRID | Dilution | Method |
|---|---|---|---|---|---|
| Anti-rabbit Cy3 | Jackson Lab (Bar Harbor, ME, USA) | Goat | AB_2 338 006 | 1:200 | IF |
| Anti-rabbit Alexa Fluor 488 | Jackson Lab | Goat | AB_2 338 052 | 1:200 | IF |
| Anti-mouse Cy3 | Jackson Lab | Goat | AB_2 338 692 | 1:200 | IF |
| Anti-mouse Alexa Fluor 488 | Jackson Lab | Goat | AB_2 338 840 | 1:200 | IF |

personnel with documented formal training and previous experience in experimental animal handling and care. All procedures were refined prior to starting the study, and the number of animals was calculated to use the smallest number of animals that was sufficient to achieve statistical significance according to prior sample power calculations.

### Origin and source of animals

We used 1- and 6-month-old $Dsg2^{mut/mut}$ male mice (Chelko et al., 2016), and littermate controls ($Dsg2^{WT/WT}$ mice). Animals were maintained in individually ventilated cages in an authorized animal facility (authorization number 175/2002A) under a 12:12 h light/dark photo-cycle, with controlled temperature and humidity, and mice had free access to water and standard rodent chow *ad libitum*. Mice were killed by cervical dislocation (in accordance with Annex IV of European Directive 2010/63/EU).

### Echocardiographic analysis

This analysis was performed in 1- and 6-month-old $Dsg2^{WT/WT}$ and $Dsg2^{mut/mut}$ mice, as described in Zaglia et al. (2014).

### Electrocardiography

This analysis was performed in 1- and 6-month-old $Dsg2^{WT/WT}$ and $Dsg2^{mut/mut}$ mice, as described in Zaglia et al. (2014).

### Immunofluorescence on murine sympathetic ganglia

Superior cervical and stellate ganglia were harvested from adult (aged 6 months) C57BL/6J male mice, fixed in 4% paraformaldehyde (PFA) [w/v in 1 × phosphate-buffered saline (PBS); Sigma-Aldrich, St Louis, MO, USA) for 1 h at room temperature, dehydrated in sucrose gradient and frozen in liquid nitrogen. Cryosections (10 μm) were obtained using a cryostat (CM1860; Leica, Wetzlar, Germany) and processed for immunofluorescence, as previously described (Zaglia et al., 2013). Yjhe rimary and secondary antibodies used in this study are listed in Tables 1 and 2. Sections were analysed using confocal microscopy (LSM900; Zeiss, Oberkochen, Germany). In total, 20 ganglia from five mice were analysed.

### Immunofluorescence on murine hearts

Hearts were harvested from 1- and 6-month-old $Dsg2^{mut/mut}$ and control $Dsg2^{wt/wt}$ mice, fixed in 1% PFA (w/v in 1X PBS; Sigma-Aldrich) and processed as previously described in Zaglia et al. (2013). Cryosections

(10 μm) were obtained using a cryostat (CM1860; Leica) and processed for immunofluorescence, as previously described (Zaglia et al., 2013). The primary and secondary antibodies used in this study are listed in Tables 1 and 2. Sections were analysed using confocal microscopy (LSM900; Zeiss).

### Assessment of sympathetic neuron density in thin heart sections

Four non-consecutive cryosections from the mid portion of the ventricles (distant 100 μm from each other) were analysed via confocal microscopy. Images of nine randomly chosen fields from the right ventricle (RV), the interventricular septum (IVS) and the left ventricle (LV) were obtained using confocal microscopy (LSM900; Zeiss) and analysed with the open-source software Fiji (Schindelin et al., 2012) to quantify the fractional myocardial area occupied by tyrosine hydroxylase positive (TH+) fibres; $n = 5$ $Dsg2^{WT/WT}$ and $n = 5$ $Dsg2^{mut/mut}$ hearts, at both 1 and 6 months of age, were analysed.

### Colorimetric map generation

Colorimetric maps were used to visualize SN distribution within sections from the mid portion of the ventricles of 6-month-old $Dsg2^{WT/WT}$ and $Dsg2^{mut/mut}$ mice. In total, five hearts per group were evaluated and, for each heart, we analysed four non-consecutive sections. Sections were stained with an antibody to TH and images acquired using fluorescence microscopy (DM6B; Leica) were composed with Fiji (Schindelin et al., 2012). Average fluorescence intensity was calculated in image bins of $150 \times 150$ pixel units, aiming to obtain a spatially resolved semi-quantitative image of neuronal distribution, and represented in pseudocolor scale ranging from blue (i.e. low density) to red (i.e. high density).

### Heart tissue clarification and whole mount immunofluorescence

To characterize the topology of SNs in the murine myocardium, in 3-D, we performed whole-mount immuno-fluorescence in optically cleared tissue blocks from different heart regions of the LV. A cardiac-optimized protocol of tissue clarification modified from Chung and Deisseroth (2013), also called the CLARITY protocol (Clear Lipid-exchanged Acrylamide-hybridized Rigid Imaging/Immunostaining/In situ hybridization-compatible Tissue-hYdrogel) was used (Giardini et al., 2021; Olianti et al., 2022). Clarified tissues are optically transparent and allow high resolution

3-D imaging and immunohistochemical/pathological analyses without tissue disassembly and the consequent loss of information as a result of sectioning artifacts. The protocol includes several subsequent steps. The heart is initially perfused with 30 mL of PBS (pH, 7.4), followed by perfusion with 24 mL of 4% PFA in PBS (Sigma-Aldrich) and fixation proceeds overnight at 4°C. After fixation, LV samples were incubated in a hydrogel solution (4% acrylamide, 0.05% bisacrylamide, 0.25% initiator AV-044, in 0.01 м PBS) for 3 days at 4°C in shaking. To favour the polymerization, oxygen was replaced by nitrogen and the tissue was maintained at 37°C for 3 h. Embedded samples were extracted from the gel and incubated in a de-lipidation solution (200 mм boric acid, 4% SDS in deionized water; pH 8.6) at 37°C with shaking for ∼3 weeks, depending on the sample size and thickness. Clarified samples were washed in PBS for 24 h with shaking at room temperature, and then in PBS, supplemented with 0.1% Triton X-100 (PBS-T 0.1X) for 24 h in shaking at room temperature. Then, they were incubated with anti-TH (dilution 1:200; AB152; Millipore, Billerica, MA, USA) in PBS-T 0.1X for 3 days at 4°C, washed in PBS for 24 h at room temperature and then incubated with the appropriate secondary antibody (anti-rabbit; dilution 1:200; AB150077; Abcam, Cambridge, UK) in PBS-T 0.1X for 2 days with shaking at room temperature. Samples were subsequently washed in PBS for 24 h at room temperature and incubated in increasing concentration of 2,2′-thiodiethanol in PBS, up to a final concentration of 68% to match the refractive index of the cleared tissue (Costantini et al., 2015). Once tissue clarification and immunostaining were completed, samples were imaged with a custom-made two-photon fluorescence microscope, equipped with a mode locked Chameleon titanium sapphire laser (120 fs pulse width, 90 MHz repetition rate; Coherent Inc., Saxonburg, PA, USA). The laser was operated at 780 nm and was coupled with a custom-made scanning system supported with a pair of galvanometric mirrors (LSKGG4/M; Thorlabs, Newton, NJ, USA), a closed-loop xy stage (U-780 PILine xy Stage System; Physik Instrumente, Karlsruhe, Germany) and a closed-loop piezoelectric stage (ND72Z2LAQ PIFOC objective scanning system, 2 mm travel range; Physik Instrumente). Two independent GaAsP photomultiplier modules (H7422; Hamamatsu Photonics, Bridgewater Township, NJ, USA) acquired the fluorescence signal. The setup was equipped with dichroic mirrors with cut-off wavelengths of 705 nm, 552 nm and 484 nm. A band-pass emission filter centred at 530 ± 55 nm was used for Alexa-Fluor 488 detection, and a filter centred at 390 ± 18 nm was used for second-harmonic generation (SHG) detection.

### Assessment of sympathetic neuron density in optically cleared heart tissues

The cardiac tissue volume was reconstructed by acquiring serial images along the *z*-plane with a custom made two-photon fluorescence microscope, which allowed 3-D mesoscale reconstruction of the SN network within its intact anatomical context at submicron resolution ($0.44 \times 0.44 \times 2$ μm, *xyz*). 3-D stacks of tissue blocks were acquired both on the epicardial ($n = 3$) and endocardial ($n = 3$) surface, in both damaged ($n = 3$) and intact ($n = 3$) regions. In total of four control and three mutant hearts (from 6-month-old mice) were analysed. To analyse cSN network topology in 3-D volumes, each stack was manually traced and segmented to reconstruct the neural network using the ImageJ plugin 'Neuro-Anatomy' (Arshadi et al., 2021; Schindelin et al., 2012). The results from tracing and segmentation were then used to calculate: (1) linear density (total length of the neuronal processes/tissue volume) and (2) neuronal density (the total number of neuronal processes/tissue volume) in control and *Dsg2^{mut/mut}* subendocardial and subepicardial regions of LV.

### Primary sympathetic neuron cultures

Postnatal day 1–day 3 (P1–P3) neonatal *Dsg2^{wt/wt}* and *Dsg2^{mut/mut}* mice were decapitated, the superior cervical and stellate ganglia were collected, and SNs were isolated as described in Prando et al. (2018). Cells were plated in laminin-coated coverslips and maintained in culture for 7 days. In a subset of experiments, we used SNs differentiated from PC12 cells, as described in Prando et al. (2018). Cultured SNs were used for immunofluorescence, morphometric and molecular analyses.

### PC12-derived sympathetic neuron infection

In a subset of experiments, SNs differentiated from PC12 cells underwent viral infection with either: Ad-Empty; Ad-sh-scramble or Ad-shDsg2-mCherry viral vectors (all from Vector Biolabs, Malvern, PA, USA), at different concentrations [multiplicity of infection (MOI) = 50, 25 and 10]. Twenty-four hours after infection, the medium was replaced and cells were analysed after 7 days.

### Immunofluorescence analysis of cultured neurons

Both primary and PC12-derived SNs were fixed in 4% PFA for 30 min at 4°C. Cells were permeabilized with PBS, supplemented with 0.1% Triton X-100 for 5 min at room temperature, and then incubated with primary antibodies as described in Prando et al. (2018). The primary and secondary antibodies used in this study are listed in Tables 1 and 2.

### Morphometric analyses in cultured sympathetic neurons

We here measured: neuronal process length, number of branch points/cell and average cell soma size by using the ImageJ plugin 'NeuroAnatomy' (Arshadi et al., 2021; Schindelin et al., 2012).

### Western blotting

*In vitro* and *ex vivo* biochemical analyses were performed as described in Zaglia et al. (2014).

### Statistical analysis

All data are expressed as the mean $\pm$ SD. Data were analysed in Prism, version 9 (GraphPad Software Inc., San Diego, CA, USA). Data were initially analysed by the Shapiro–Wilk test to determine their distribution. In datasets showing normal distribution, an unpaired *t* test or unpaired *t* test with Welch's correction (comparison between two groups) or one-way analysis of variance (ANOVA) (comparison between three or more groups) were applied. For one-way ANOVA, depending on the result of SD equality (by a Brown–Forsythe test), ordinary or Brown–Forsythe and Welch ANOVA tests were performed. In non-normally distributed datasets, non-parametric tests were used. Multiple comparisons were corrected as suggested by Prism. $P < 0.05$ was considered statistically significant.

## Results

### Sympathetic neurons express desmosomal proteins

In this study, we first assessed the expression of desmosomal proteins by SNs innervating the heart. To this aim, superior cervical and stellate ganglia were isolated from normal adult C57BL/6J male mice and used for biochemical assays, which demonstrated the expression, by the cell soma of heart innervating neurons, of TH (i.e. the rate-limiting enzyme in sympathetic neuronal catecholamine synthesis), thus confirming their sympathetic nature, and desmosomal proteins: plakoglobin (JUP), plakophilin-2 (PKP2), desmoglein-2 (DSG2) and desmoplakin (DSP) (Fig. 1*A*). Among these, we turned our attention to DSG2 because *DSG2* variants cause a left-dominant form of AC with marked myocardial remodelling and fibrosis (Chelko et al., 2016; Chua et al., 2023; Pilichou et al., 2006). The expression of DSG2 by cSNs was further confirmed by confocal immunofluorescence in murine and, notably, human

stellate ganglia sections (Figs 1*B,C* and 2). To assess subcellular localization of DSG2 in SNs, primary cells from cardiac sympathetic ganglia were isolated and cultured as described in Prando et al. (2018). Confocal immunofluorescence evidenced that DSG2-immunoreactivity was detectable in both the cell soma and axonal processes (Fig. 3*A,B*). Nuclear DSG2 expression was confirmed in isolated stellate ganglia (Fig. 3*C*). Similar results were obtained *in vitro* in SNs derived from PC12 cells, an immortalized catecholamine-expressing neuronal cell line, thus indicating that DSG2 is endogenously expressed in SNs independent from interaction with the innervated target organ (Fig. 3*D,E*). These results imply that, in the DSG2-mutation carriers, SNs would harbour the mutant protein and may thus comprise additional cells affected in AC.

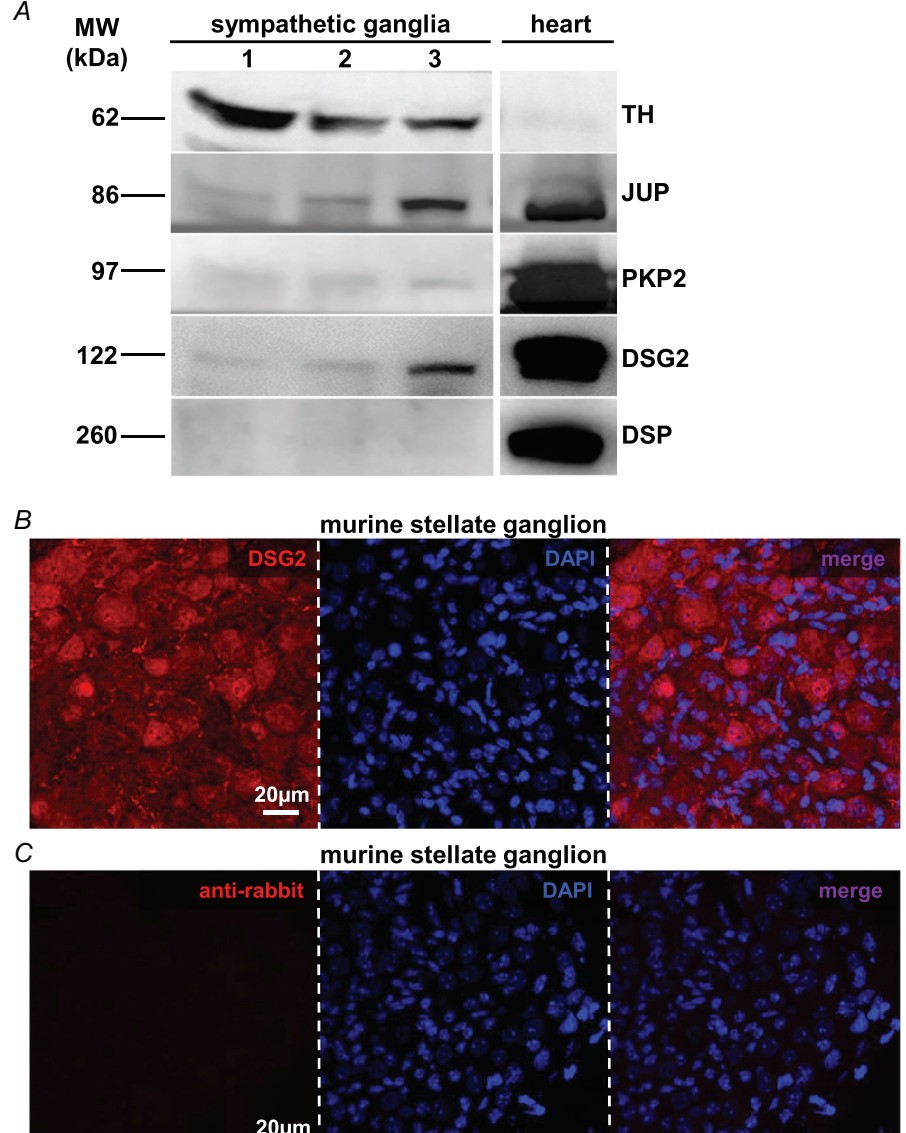

**Figure 1. Desmoglein-2 is expressed by murine cardiac sympathetic neurons**
*A*, western blotting on protein extracts from superior cervical and stellate ganglia of adult (6 months) C57BL/6J male mice to assess the expression of desmosomal proteins (JUP, plakoglobin; PKP2, plakophilin-2; DSG2, desmoglien-2; DSP, desmoplakin). Tyrosine hydroxylase (TH) was used to ensure equal protein loading. Heart extracts were used as positive controls. MW, molecular weight. The numbers indicate different animals from which sympathetic ganglia were harvested. In total, five mice were analysed. Experiments were repeated three times. Part of the image in Fig. 4*A* is shown here for the purpose of data normalization. *B* and *C*, confocal immunofluorescence on murine stellate ganglia sections, stained with anti-desmoglein2 (*B*, DSG2, red signal) or anti-rabbit (*C*, red signal). Nuclei were counterstained with DAPI (blue signal). The right panels in (*B*) and (*C*) represent merged images. In total, 20 ganglia from five mice were analysed. [Colour figure can be viewed at wileyonlinelibrary.com]

## DSG2 downregulation affects sympathetic neurons

To evaluate whether AC-linked DSG2 downregulation affects cSNs, we analysed cultured SNs isolated from the superior cervical/stellate ganglia of neonatal *Dsg2^mut/mut* mice, in which the exons 4 and 5 of murine *Dsg2* gene were excised, leading to nonsense-mediated mRNA decay and complete loss of DSG2 protein in DSG2-expressing cells (Chelko et al., 2016), including SNs (Fig. 4*A*). Isolated neurons were maintained in culture for 7 days to ensure their *in vitro* maturation towards the differentiated SN phenotype (Prando et al., 2018). Confocal immunofluorescence using the specific sympathetic marker (TH) and the synaptosomal-associated protein 25 (SNAP-25) revealed profound morphological abnormalities in AC neurons (Fig. 4*B*). Indeed, the axonal processes of *Dsg2^mut/mut* cSNs were shorter and less ramified than control (*Dsg2^wt/wt*) ones (Fig. 4*C,D*). No significant changes in the size of cell soma were observed (Fig. 4*E*). Yet upon further inspection, axonal processes of *Dsg2^mut/mut* cells were thicker and showed reduced number of irregularly distributed varicosities (Fig. 4*B*). To determine whether DSG2 downregulation has a primary role in affecting SNs, we analysed neuronally differentiated PC12 cells, infected with a viral vector encoding a *Dsg2* sh-RNA fused with the fluorescent reporter mCherry, at different MOIs. Ad-Empty and Ad-sh-scramble vectors were used as controls. Western blotting analysis confirmed the expected effect of viral infection, as DSG2 protein level was reduced by ∼75% in cells infected with Ad-*shDsg2*, already at low MOI (Fig. 5*A*). DSG2 level was unchanged in cells infected with either Ad-Empty or Ad-sh-scramble vectors (Fig. 5*B*). Infection with Ad-*shDsg2* vector led to reduction in SN density, compared to Ad-Empty and Ad-sh-scramble vectors (Fig. 5*C*) and, notably, Ad-*shDsg2* infection replicated key cellular phenotypes from *Dsg2^mut/mut* primary neurons, such as decreased neurite length, branching and irregular distribution of varicosities (Fig. 6). These results support the causal role of DSG2 downregulation in affecting SNs and prompted further investigation of cardiac innervation in the intact *Dsg2^mut/mut* AC hearts.

## Sympathetic innervation is altered in *Dsg2^mut/mut* hearts

Confocal immunofluorescence was used to compare cardiac sympathetic innervation in male *Dsg2^mut/mut* mice at 1 month (early disease stage prior to heart damage, contractile dysfunction and arrhythmias appearance) and 6 months (advanced disease stage with extensive heart remodelling, decrease contractile performance and

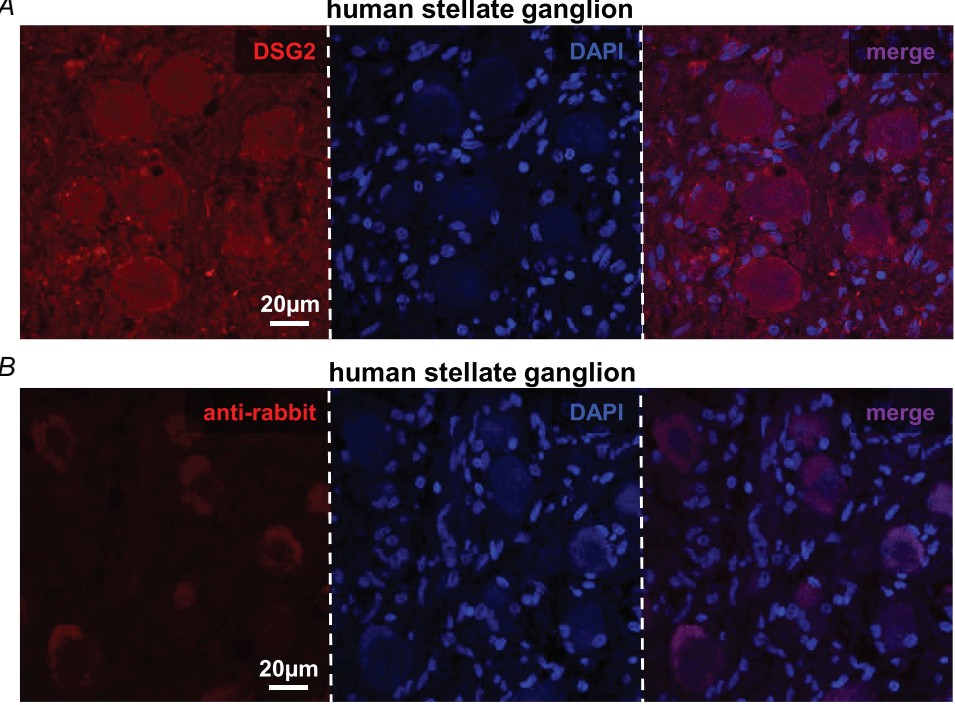

**Figure 2. Desmoglein-2 is expressed by human cardiac sympathetic neurons**
*A* and *B*, confocal immunofluorescence on autoptic human stellate ganglia sections, stained with anti-desmoglein2 (*A*, DSG2, red signal) or anti-rabbit (*B*, red signal). Nuclei were counterstained with DAPI (blue signal). The right panels in (*B*) and (*C*) represent merged images. Post-mortem stellate ganglia samples from two patients were analysed. [Colour figure can be viewed at wileyonlinelibrary.com]

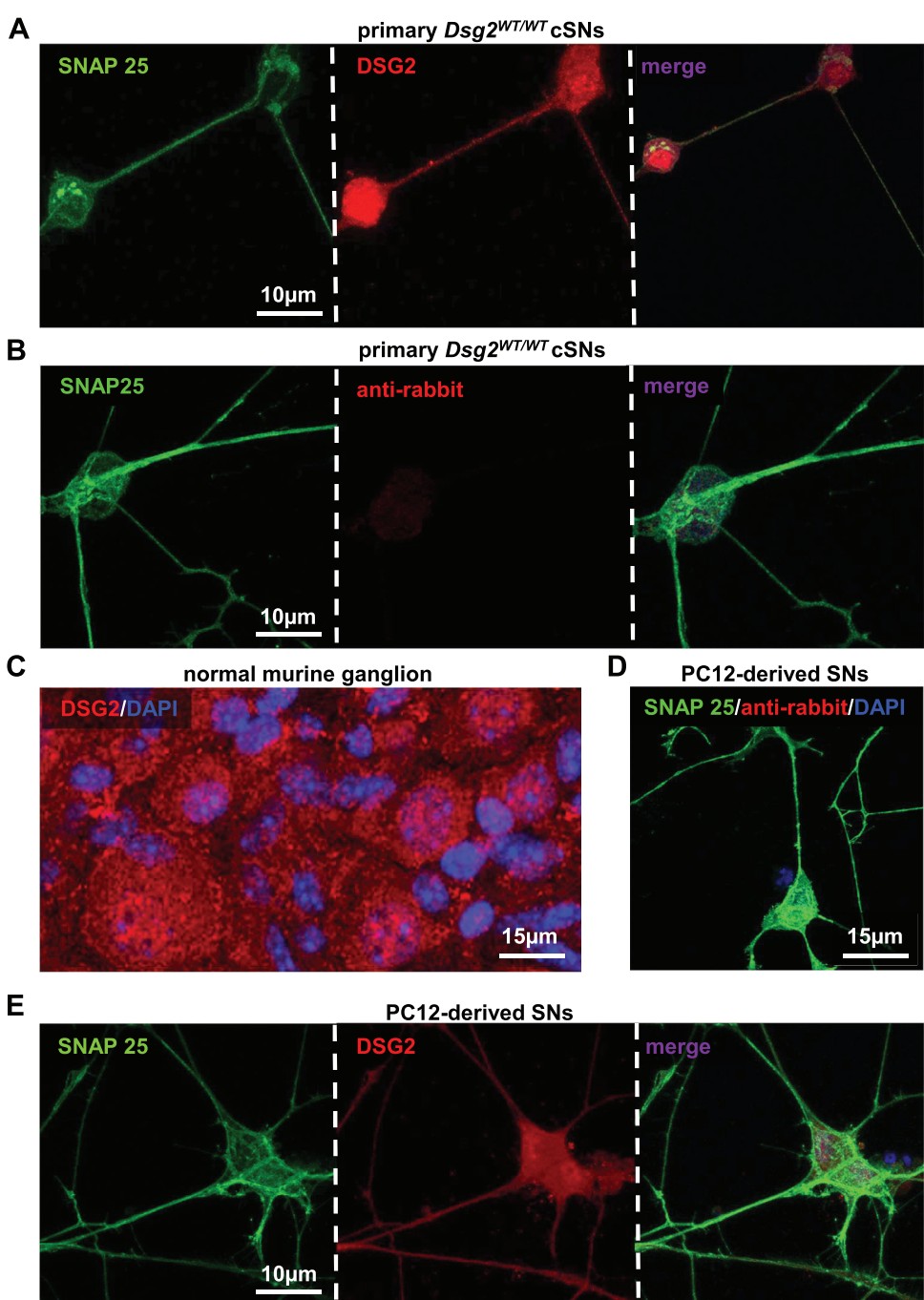

**Figure 3. Desmoglein-2 is expressed in the cytoplasm and nucleus of sympathetic neurons**

*A* and *B*, confocal immunofluorescence on cultured cardiac sympathetic neurons (cSNs) isolated from the superior cervical/stellate ganglia of adult normal mice, co-stained with anti-SNAP-25 (green signal, left) together with: anti-DSG2 (*A*), red signal, middle) or anti-rabbit (*B*), red signal, middle). Nuclei were counterstained with DAPI (blue signal). The right panels in (*A*) and (*B*) show merged images. Images are representative of three independent experiments. In each experiment, we used 12 ganglia from three *Dsg2^{wt/wt}* mice, and we analysed at least 30 cells. *C*, high magnification of the image shown in Fig. 1*B*. The panel shows the coexistence of cells expressing nuclear DSG2 with cells negative for DSG2 expression, thus underlying antibody specificity. In total, 20 ganglia from five mice were analysed. *D*, confocal immunofluorescence on cultured PC12-derived SNs, co-stained with antibodies to SNAP-25 (green signal) and anti-rabbit (red signal). Nuclei were counterstained with DAPI (blue signal). *E*, confocal immunofluorescence on cultured PC12-derived SNs, co-stained with antibodies to SNAP-25 (green signal, left) and anti-DSG2 (red signal, middle). Nuclei were counterstained with DAPI (blue signal). The right panel shows the merged image. Images in (*D*) and (*E*) are representative of three independent experiments. In each experiment, we analysed at least 30 cells per experiment. [Colour figure can be viewed at wileyonlinelibrary.com]

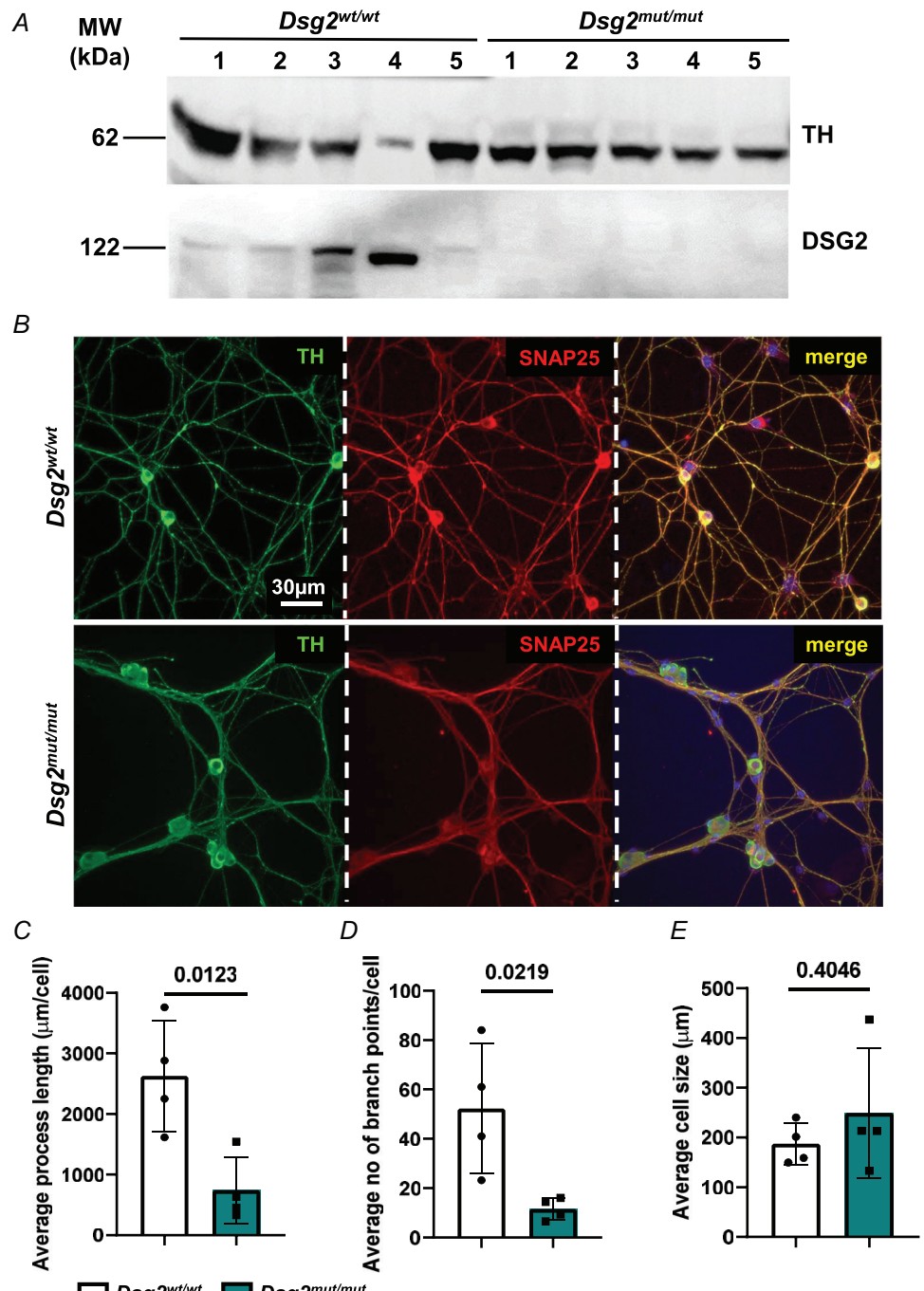

**Figure 4. Cardiac sympathetic neurons from *Dsg2*<sup>mut/mut</sup> mice show morphologic abnormalities**

*A*, western blotting on protein extracts from superior cervical/stellate ganglia from adult *Dsg2*<sup>mut/mut</sup> male mice and littermates (*Dsg2*<sup>w/wt</sup>) to assess DSG2 protein content. Tyrosine hydroxylase (TH) was used to ensure equal protein loading. MW, molecular weight. The numbers indicate the different biological replicates analysed. The image is partly replicated in Fig. 1*A* for the purpose of data normalization. The image is representative of three independent experiments. We analysed a total of 20 ganglia from five control *vs.* five AC mice. *B*, confocal immuno-fluorescence of cultured primary cardiac sympathetic neurons (cSNs) isolated from the superior cervical/stellate ganglia of neonatal *Dsg2*<sup>w/wt</sup> and *Dsg2*<sup>mut/mut</sup> mice. Cells were co-stained with antibodies to TH (left) and SNAP-25 (middle). Nuclei were counterstained with DAPI. The right panel shows the merged image. *C* and *E*, quantification of the average length of neuronal processes (*C*), number of neuronal ramifications (*D*) and size of cell soma (*E*) in cultured control *vs. Dsg2*<sup>mut/mut</sup> cSNs. In total, 20 ganglia from five *Dsg2*<sup>wt/wt</sup> and five *Dsg2*<sup>mut/mut</sup> mice were used. Bars represent the SD. Differences among groups were evaluated using unpaired *t* test. Each value represents quantifications performed in 4 randomly chosen fields; *n* = 17 *Dsg2*<sup>wt/wt</sup> cells *vs. n* = 25 *Dsg2*<sup>mut/mut</sup> SNs. [Colour figure can be viewed at wileyonlinelibrary.com]

increased arrhythmic incidence) (Chelko et al., 2016) (Fig. 7). At both ages, we observed marked differences in morphology, density and topology of cSNs in AC hearts compared to controls. As shown in Fig. 8*A*, SN processes appeared thicker in *Dsg2^{mut/mut}* hearts and showed irregular enlargements, which were more evident in hearts from adult mice. Quantitative analysis in thin tissue slices evaluated SN density as percentage of myocardial area occupied by TH-positive fibres, in randomly sampled fields irrespective of tissue remodelling (Fig. 8*B*). SN density appeared higher in the RV wall and IVS of 1-month *Dsg2^{mut/mut}* hearts and further increased in all regions (including the LV) in 6-month hearts. In the latter hearts, which are characterized by diffused foci of myocardial damage, quantification of neuronal density

across random areas of the heart yielded uneven values, which reflected the elevated variance of the sampled density distribution, as shown in the plot in Fig. 8*C*. A colour-coded composite image was generated by binning the ventricular section image with a shrink factor of 150 in both *x* and *y* dimensions and represents the average neuronal density across the section, encoded as in the corresponding colour bar (from blue = low SN density to red/purple = high SN density) (Fig. 9). The relative pseudo-colours emphasize the expected transmural pattern of sympathetic innervation in the normal heart, characterized by the prevailing appearance of neurons in the subepicardial layers, with a well-determined sub-epicardial/subendocardial ratio (Fig. 9*A*). Remarkably, SN distribution was grossly altered in *Dsg2^{mut/mut}* hearts,

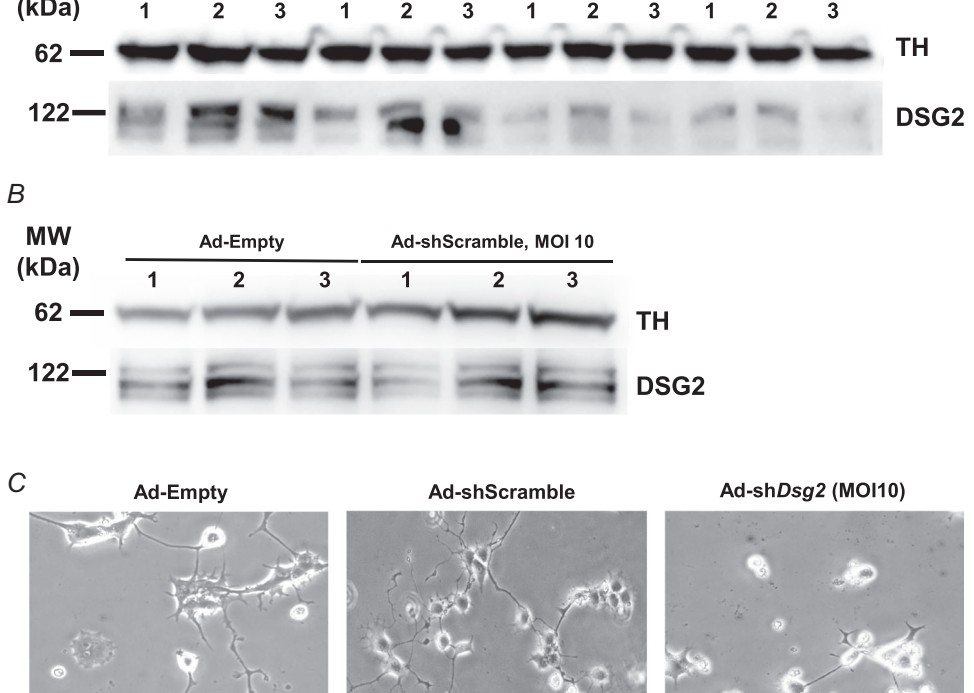

**Figure 5. Cultured sympathetic neuron viral infection efficiently down-regulates desmoglein-2**
*A*, western blotting on protein extracts of PC12-derived sympathetic neurons (SNs) infected with a viral vector encoding sh-*Dsg2* fused with the fluorescent protein m-cherry. PC12-derived SNs infected with an Empty viral vector (Ad-Empty) were used as control. Tyrosine hydroxylase (TH) was evaluated to ensure equal protein loading. *B*, western blotting on protein extracts of PC12-derived SNs infected with either an Empty (Ad-Empty) or an Ad-sh-scramble viral vector. TH was evaluated to ensure equal protein loading. In (*A*) and (*B*) images are representative of three independent experiments. MW, molecular weight. *C*, bright filed image of PC12-derived SNs analysed 7 days upon infection with the following viral vectors: Ad-Empty (left panel), Ad-sh-scramble (middle) or Ad-sh*Dsg2* (right). Images are representative of three independent experiments.

which showed the coexistence of hyper-innervated regions (identified by yellow-red colours), frequently found in subepicardial areas, irregularly interspersed to areas of greatly reduced innervation density (green-blue colours) (Fig. 9*B*). We were thus prompted to finely characterize the sympathetic network topology in 3-D, as well as its relationship with local myocardial remodelling.

## The sympathetic neuron network is profoundly affected in *Dsg2*$^{mut/mut}$ hearts

Whole-mount immunofluorescence with anti-TH was performed in clarified blocks from either the subepicardial or subendocardial sides of the LV and were then imaged with a two-photon fluorescence microscope

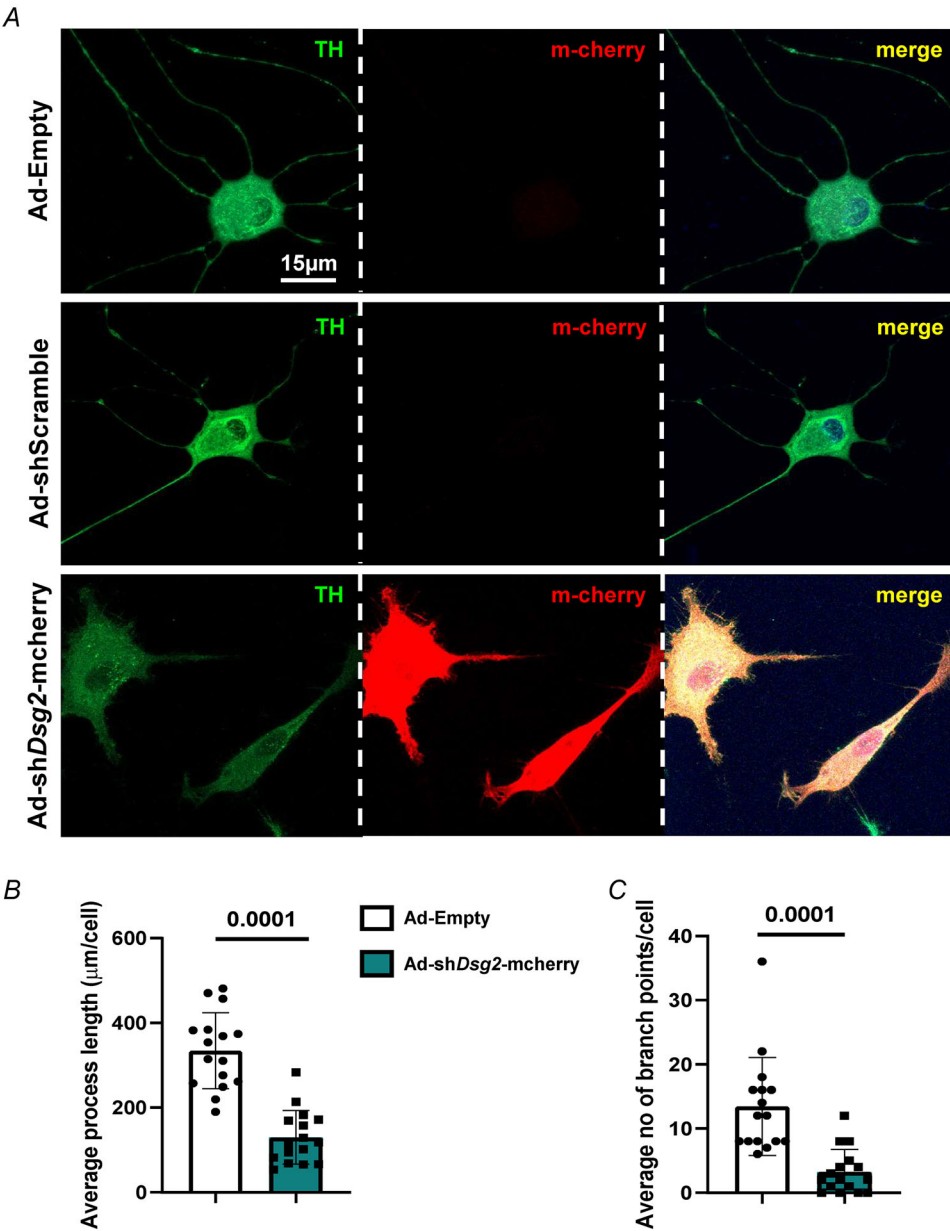

**Figure 6. Desmoglein-2 downregulation primarily affects sympathetic neurons**

*A*, confocal immunofluorescence on PC12-derived SNs infected with either Ad-Empty, Ad-sh-scramble or Ad-sh*Dsg2*-mcherry viral vectors. Cells were stained with antibodies to TH (left). The red fluorescence of m-cherry was used to identify cells infected with Ad-sh*Dsg2* viral vector (middle). Nuclei were counterstained with DAPI. The right panel shows the merge image. Images are representative of three independent experiments. *B* and *C*, quantification of the average length of neuronal processes (*B*) and number of neuronal ramifications (*C*), in cultured SNs infected with either control or Ad-sh*Dsg2* viral vectors. Bars represent the SD. Differences among groups were evaluated using an unpaired *t* test (*B*) and a Mann–Whitney test (*C*). *n* = 20 *Dsg2*$^{w/twt}$ *vs. n* = 21 *Dsg2*$^{mut/mut}$ SNs. [Colour figure can be viewed at wileyonlinelibrary.com]

(Olianti et al., 2020). Each cardiac tissue volume was reconstructed by overlapping *z*-images, allowing for a 3-D mesoscale reconstruction of the SN network within its anatomical context at submicron resolution. Image series were classified according to the proximity to remodelled areas, in 'damaged' or 'intact' myocardium. Figure 10*A* highlights the high complexity of the cSN network found in the 'intact' heart regions of *Dsg2^{wt/wt}* hearts, showing the preserved subepicardial/subendocardial neuronal density gradient that was, strikingly, amplified in AC hearts as a result of the massive hyperinnervation of the subepicardial regions. Such an increase in subepicardial SN density was confirmed by the quantitative network analysis in the 3-D volumes, showing that both the number and total length

of neuronal processes were significantly increased in the *Dsg2^{mut/mut}* hearts (Figs 10*B* and 11). The combination of two photon microscopy immunofluorescence with SHG imaging allowed inspection of the SN network in the 'damaged' areas, which were hallmarked by SHG signal consistent with collagen deposition (Vanzi et al., 2012). This method revealed that lesional foci in both subepicardial and subendocardial regions were heterogeneously innervated by grossly disarranged neuronal processes, which were evident at the sides and within the myocardial scars (Fig. 10*C*). Remarkably the number and length of neuronal processes in the 'damaged' sites were reduced to levels comparable to those of age- and sex-matched *Dsg2^{wt/wt}* hearts (Fig. 11).

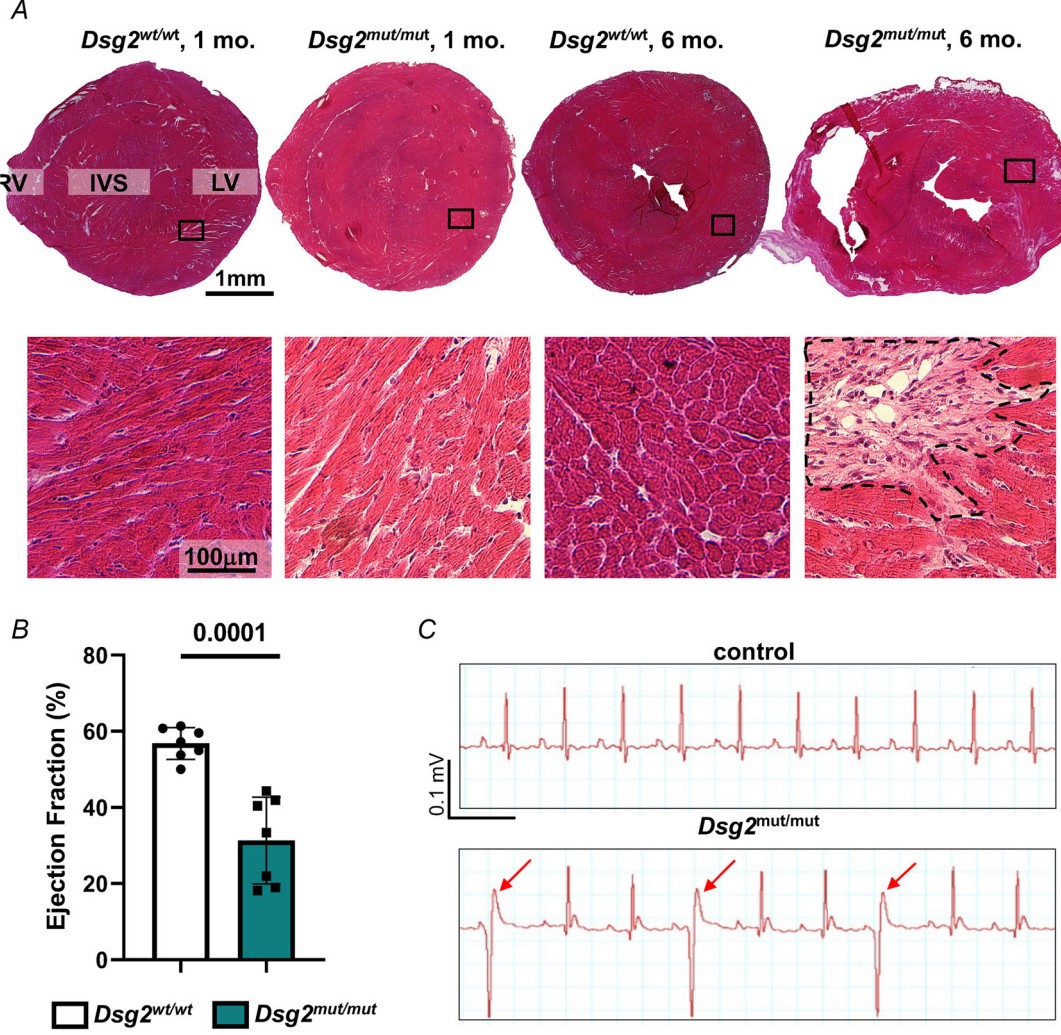

**Figure 7. Structural and functional cardiac phenotyping of *Dsg2^{mut/mut}* mice**
*A*, haematoxylin and eosin staining of ventricular sections from the mid portion of the ventricles from 1- *vs.* 6-month-old *Dsg2^{WT/WT}* or *Dsg2^{mut/mut}* mice. The bottom panels show the high magnifications of the black boxes in the LV of composite images. RV, right ventricle; IVS, interventricular septum; LV, left ventricle. We analysed six hearts for each group. *B* and *C*, echocardiographic (*B*) and electrocardiographic (*C*) analysis of 1- *vs.* 6-month-old *Dsg2^{WT/WT}* or *Dsg2^{mut/mut}* mice. Bars in (*B*) represent the SD. Differences among groups were evaluated using unpaired *t* test; *n* = 7 mice per group. [Colour figure can be viewed at wileyonlinelibrary.com]

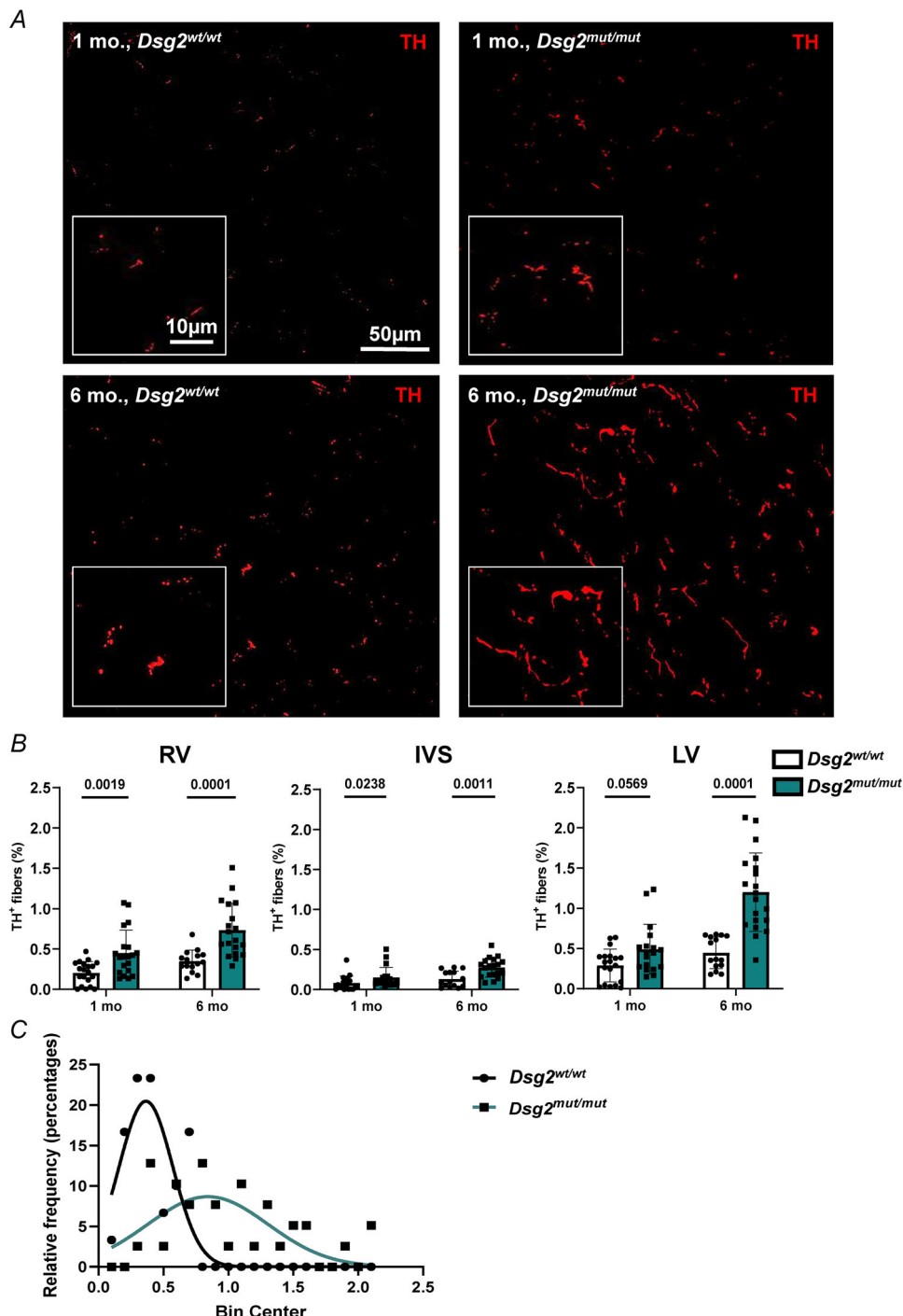

**Figure 8. Increased sympathetic innervation in *Dsg2^{mut/mut}* hearts**
*A*, confocal immunofluorescence on heart sections from the mid portion of the ventricles of young (1 month) and adult (6 months) control and *Dsg2^{mut/mut}* mice, stained with anti-tyrosine hydroxylase (TH, red signal). *B*, quantification of the fraction of myocardial area occupied by TH positive fibres in heart sections from the mid portion of the ventricles of *Dsg2^{wt/wt}* vs. *Dsg2^{mut/mut}* mice at 1 and 6 months of age. SN density was assessed in the right ventricle (RV), interventricular septum (IVS) and in the left ventricle (LV). Bars represent the SD. Differences among groups were assessed using a Mann–Whitney test. Each value represents the mean percentage of TH-positive fibres/tissue area calculated in three randomly chosen fields from four non-consecutive cryosections; *n* = 5 hearts for each group. *C*, distribution of SN density in randomly sampled regions in hearts from 6-month-old *Dsg2^{wt/wt}* and *Dsg2^{mut/mut}* mice. The best-fitting Gaussian functions obtained through non-linear regression analysis show the distribution of the SN density across the heart. [Colour figure can be viewed at wileyonlinelibrary.com]

## Discussion

By combining *ex vivo* analyses on murine and human stellate ganglia samples, multiphoton imaging of clarified heart blocks and *in vitro* assays in cultured neurons demonstrates that AC-linked *Dsg2* mutations lead to significant effects on cardiac autonomic innervation. Our results reveal an unrecognized feature of AC hearts, which, in addition to the undoubted role of mutant cardiomyocytes, is potentially involved in the mechanism of increased arrhythmogenic propensity hallmarking the disease.

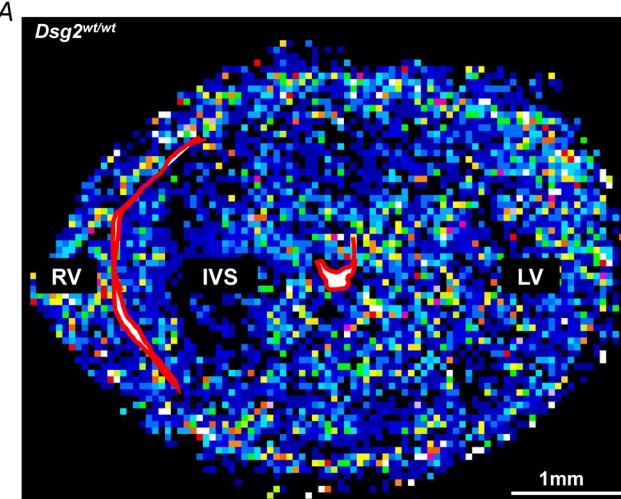

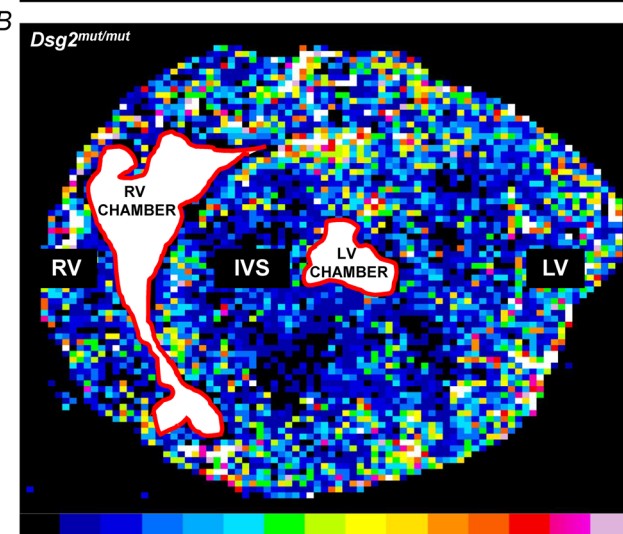

**Figure 9. Sympathetic neuron topology is altered in the left ventricle of adult *Dsg2^mut/mut* hearts**

*A* and *B*, pseudo-colour scale maps obtained from anti-tyrosine hydroxylase (TH) stained ventricular sections of adult (6 months) control (*A*) *vs. Dsg2^mut/mut* (*B*) male mice. The red line delineates ventricular chambers. Colours range from blue to yellow/red, indicating lower or higher TH fluorescence intensity, respectively. RV, right ventricle; IVS, interventricular septum; LV, left ventricle; *n* = 5 hearts for each group. [Colour figure can be viewed at wileyonlinelibrary.com]

AC is a familial cardiac disease and represents one of the main causes of stress-related arrhythmic sudden death, predominantly in the young and athletes (Corrado et al., 2001; Corrado et al., 2006; Mazzanti et al., 2016; Pilichou et al., 2016; Thiene et al., 1988; van der Voorn et al., 2020). AC is caused, in ∼50% of genetically diagnosed cases, by mutations in genes encoding desmosomal proteins, which, in cardiomyocytes, organize into desmosomes and allow the heart to sustain the mechanical stress of contractions throughout the entire lifespan (Basso et al., 2009; Gerull et al., 2004; Giuliodori et al., 2018; McKoy et al., 2000; Pilichou et al., 2006; Rampazzo et al., 2002; Syrris et al., 2006). In the AC heart, cardiomyocyte death and tissue inflammation lead to myocardial fibro-fatty remodelling, which was originally detected in the right ventricle and, as such, the disease was named arrhythmogenic right ventricular cardiomyopathy. The spectrum of diseases within such diagnostic designation has, however, grown far beyond and now includes forms characterized by biventricular or left-dominant remodelling under the comprehensive definition of AC. The fibro-fatty lesions of the AC myocardium compromise contractile performance (sometimes leading to heart failure) and cardiac electrical activity (generating a pro-arrhythmogenic substrate) (Corrado et al., 2005; Corrado et al., 2017; di Gioia et al., 2016; Migliore et al., 2013; Zorzi et al., 2016). AC is indeed characterized by life-threatening ventricular arrhythmias, which, in several studies in animal and human models, have also been attributed to the effects of AC-linked mutations on cardiomyocyte electrophysiology. In detail, it has been demonstrated that destabilization of desmosomes leads to electrical remodelling, caused by alterations in gap junction organization and connexin expression (Chevalier et al., 2021; Gehmlich et al., 2011; Oxford et al., 2007; Reisqs et al., 2023; Rizzo et al., 2012; Stevens et al., Cells, 2022). Perturbation of the desmosomal structure can also lead to the loss of voltage-gated sodium channels (Na$_v$1.5), which often occurs in hearts with minor or undetectable structural lesions (Cerrone & Delmar., 2014; Reisqs et al., 2023; Rizzo et al., 2012; Zaklyazminskaya & Dzemeshkevich, 2016). In addition, mutations in desmosomal proteins affect cardiomyocyte expression of genes controlling $Ca^{2+}$ dynamics, causing arrhythmogenic alteration in $Ca^{2+}$ handling (Cerrone et al., 2017; Reisqs et al., 2023 Vallverdù-Prats et al., 2023; Tiso et al., 2001; van der Zwaag et al., 2012). Such changes have also been described in the context of physical (i.e. exercise) or emotional (i.e. psychosocial stress) stresses, with both conditions triggering life-threatening arrhythmias and independently accelerating AC progression (Agrimi et al., 2020; Asimaki et al., 2015; Chelko et al., 2016; Corrado & Zorzi., 2015; Corrado et al., 2003; Corrado et al., 2017; Corrado et al., 2019; Martherus et al., 2016; Saberniak et al., 2014; Wichter et al., 2000).

Notably, exercise and emotions are well-known activators of the sympathetic nervous system, for which activation may favour the onset of re-entry mechanisms around the fibro-fatty myocardial areas (Gardner et al., 2016;

Habecker et al., 2025; Herring et al 2019; Ripplinger et al., 2016; Zaglia & Mongillo., 2017).

The evidence that unbalanced adrenergic stimulation is potently arrhythmogenic, and that endurance training

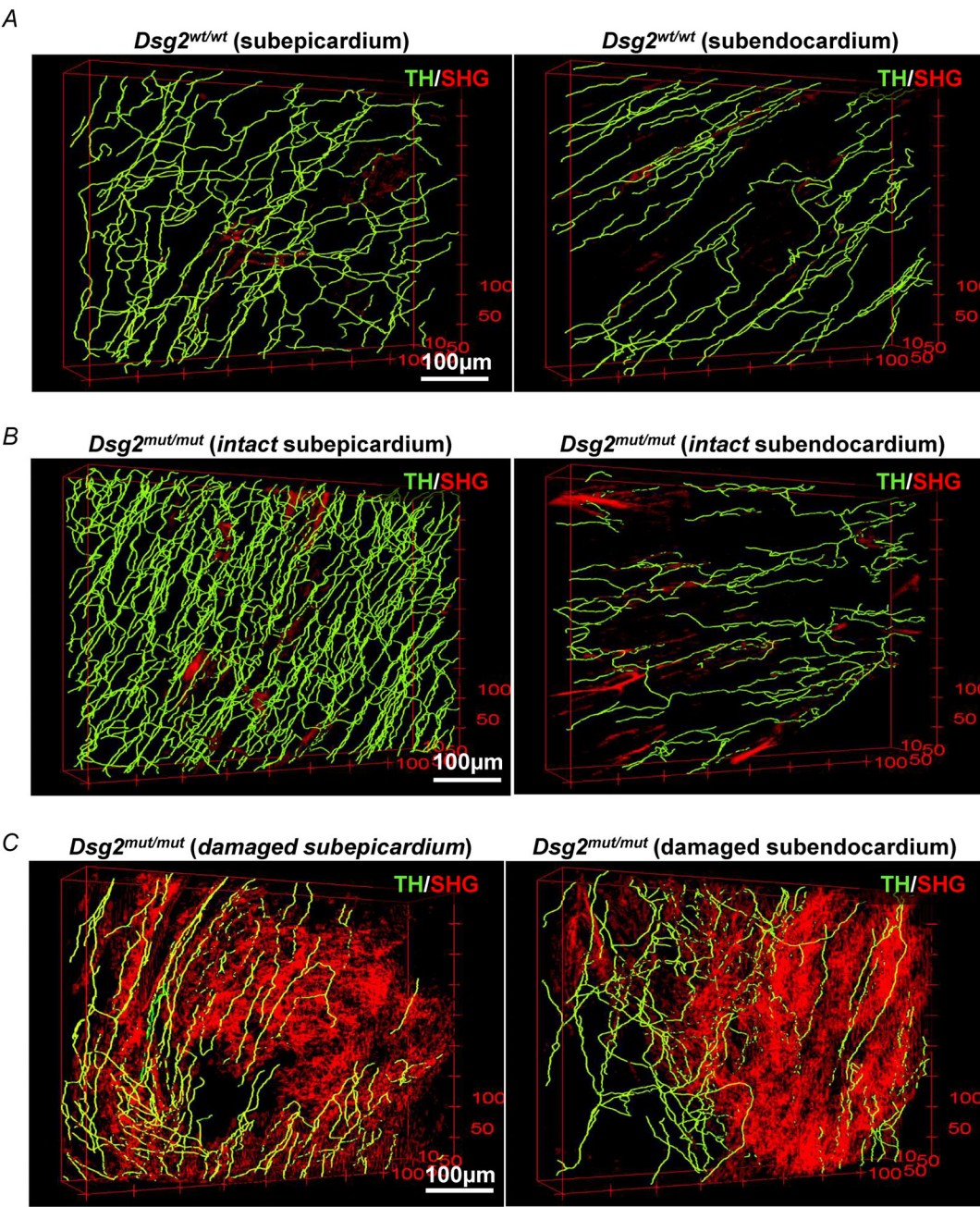

**Figure 10. Sympathetic neuron network is profoundly altered in *Dsg2^mut/mut* hearts**
*A–C*, representative reconstruction of the sympathetic neuron network in 450 × 450 × 450 μm (respectively *x*, *y*, and *z*) subepicardial (left) and subendocardial (right) regions of whole-mounted tissue-clarified heart blocks of adult (6 months) male mice, (*A*) *Dsg2^wt/wt* hearts, (*B*) *Dsg2^mut/mut* hearts in the intact areas, and (*C*) *Dsg2^mut/mut* hearts in the damaged areas, stained with anti-tyrosine hydroxylase (TH). Samples were imaged and analysed with a custom-made two-photon fluorescence microscope, combined to second harmonic generation (SHG) imaging to detect the signal generated by collagen (red signal) allowing to distinguish 'intact' from 'damaged' areas in AC heart blocks. [Colour figure can be viewed at wileyonlinelibrary.com]

and/or emotional stress accelerate disease progression (Corrado et al., 1997; Corrado et al., 2019; Corrado et al., 2020; Denis et al., 2014), together with clinical suggestion that SNs may be functionally altered in AC hearts (Myles et al., 2012., Schinner et al., 2017; Wichter et al., 2000), supports the involvement of cSNs in AC development and manifestations. Interestingly, recent evidence shows that sympathetic hyperinnervation occurs in a stress-related arrhythmic syndrome model, namely catecholaminergic polymorphic ventricular tachycardia, which shares the stress-dependency of arrhythmias with AC, but none of the structural remodelling features (M O' reilly et al., 2024).

Current research has approached 'AC as a disease of cardiomyocytes' because they carry desmosomes. Consequently, almost all preclinical models of AC were based on the overexpression or downregulation of the disease-mutant gene selectively in cardiomyocytes and, unsurprisingly, only partially recapitulated the clinical phenotype (Asimaki et al., 2014; Cerrone et al., 2012; Kam et al., 2018; Li et al., 2011; Lyon et al., 2014; Pilichou et al., 2011). Interestingly, desmosomal proteins are expressed by all cardiac cell types, including intrinsic (i.e. cardiomyocytes, resident inflammatory cells, endothelial and smooth muscle cells) and extrinsic (i.e. autonomic neurons) cell populations, all of which may participate in AC development (Scalco et al., 2021; Sommariva et al., 2016; Stadiotti et al., 2021). We thus approached AC as a multicellular disorder affecting cross-talk among the different cardiac cell types and investigated whether sympathetic innervation is altered in AC hearts. To this aim, we exploited a recently developed AC murine model

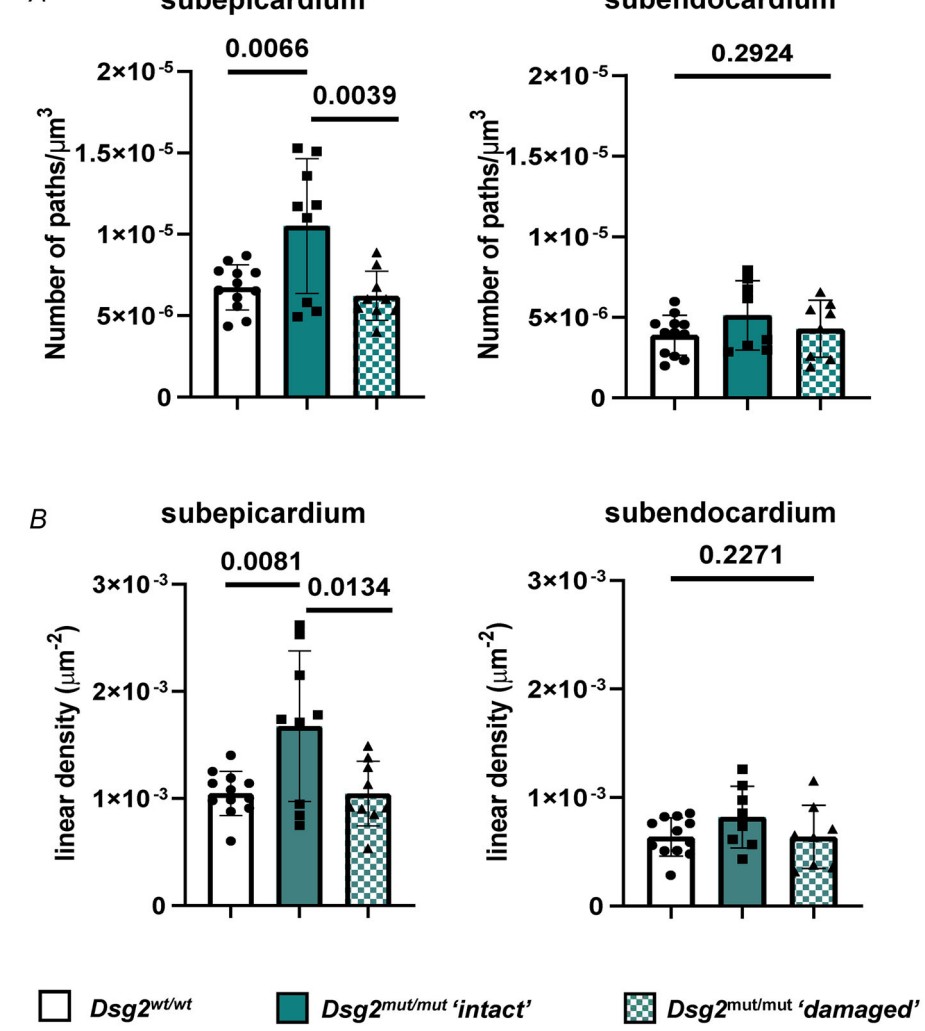

**Figure 11. Quantitative analysis of the sympathetic neuron network in *Dsg2^mut/mut* hearts**
*A* and *B*, evaluation of SN density (*A*) and linear density (*B*) in subepicardial and subendocardial regions of adult (6 months) *Dsg2^wt/wt* and *Dsg2^mut/mut* male mice. Bars represent the SD. Difference among groups was determined using one-way ANOVA; *n* = 4 *Dsg2^wt/wt* *vs.* *n* = 3 *Dsg2^mut/mut* mice. [Colour figure can be viewed at wileyonlinelibrary.com]

that carries the excision of exons 4 and 5 in the *Dsg*2 gene (*Dsg2^{mut/mut}* mice) and mimics the germline expression of the disease-linked gene variants occurring in humans (Chelko et al., 2016).

Sympathetic neurons innervating the heart originate in paravertebral ganglia, which extend long axons to the heart and ramify within the myocardium with a pearl-necklace morphology, characterized by regularly distributed varicosities representing the neurotransmitter releasing sites. Cardiac SN processes distribute in the heart with precise species-specific topology established in the early postnatal weeks as a result of the balanced effect of chemoattract (i.e. neurotrophins) and chemorepellent (i.e. SEMA-2a) factors released by cardiomyocytes and vascular cells (Franzoso et al., 2016; Habecker et al., 2016; Ieda et al., 2007; Kimura et al., 2012; Larsen et al., 2016; Pianca et al., 2019; Prando et al., 2018; Zaglia et al., 2013). Accordingly, in the murine ventricles, the subepicardium is more densely innervated than the sub-epicardial region, and such a topology is maintained in the adult heart as a result of unremitted feeding of neurons by cardiomyocyte-derived neurotrophins (Franzoso et al., 2016; Pianca et al., 2019). On this basis, we evaluated sympathetic innervation in thin sections of *Dsg2^{mut/mut}* hearts with respect to the development of AC lesions. Hearts were thus analysed at early phase (1 month), characterized by absence of damage, and at an advanced disease stage (6 months), when the heart has several lesions, contractile dysfunction and increased arrhythmia incidence. Quantitative immunofluorescence demonstrated that innervation of *Dsg2^{mut/mut}* mice is globally enhanced with increased axonal sprouting and irregular morphology and distribution of neuronal varicosities. Such alterations were evident already at the early disease stage and increased during disease progression, resulting in significant changes in the physiological SN topology because of reduced transmural gradient of SN density and the co-existence of hypo- and hyper- innervated myocardial areas. Such aspects of cardiac autonomic neuron anatomy were analysed in unprecedented detail in the present study by exploiting multiphoton confocal immunofluorescence in tissue clarified heart blocks. 3-D reconstruction of the neuronal network revealed that, in the outer layers of the *Dsg2^{mut/mut}* LV, massive hyperinnervation occurs in the 'intact' myocardial areas, far from the AC lesions, and this accounts for most of the increased SN density observed in thin sections. Remarkably, the use of such an advanced imaging method reveals that SNs may heterogeneously distribute around and into the fibrotic areas, which is a finding consistent with the inspection of autoptic heart samples from AC patients, as suggested by Stadiotti et al. (2021). The finding that abnormalities in the sympathetic innervation of AC hearts appear before myocardial remodelling is detectable suggests that this may be an independent event preceding and participating in myocardial remodelling, and that SNs may be directly affected by AC-linked *Dsg2* mutations. In further support of this hypothesis, cSNs isolated from superior cervical/stellate ganglia of *Dsg2^{mut/mut}* neonatal mice, prior to the establishment of cardiac innervation, show compromised axonal sprouting, process development and irregular distribution of varicosities. Consistently, virus-assisted DSG2 downregulation replicated in PC12-derived SNs the phenotypic alterations displayed by *Dsg2^{mut/mut}* primary neurons, corroborating that AC-linked *Dsg2* variants may primarily affect SNs. Taking together the experimental evidence that DSG2 has a primary role in SN biology, the different phenotypes of SNs in the *in vitro vs*. the *in vivo* contexts and the disease stage-dependency of sympathetic neuropathology of the AC hearts, these suggest that altered cardiac sympathetic innervation may result from the combination of cell-autonomous (i.e. primary effect of DSG2 mutation in SNs) and context-dependent factors (i.e. aberrant release of chemorepellent/chemoattractant factors from mutant AC myocardium) implicated in myocardial remodelling. This latter point is in line with increasing evidence where primary myocardial damage may secondarily affect cSNs (Dokshokova et al., 2022; Habecker et al., 2016).

Taken together, our results favour the concept that AC is a disease of multiple cell types, which, in addition to cardiomyoyctes (Basso et al., 2009; Basso et al., 2011; Gerull et al., 2004; Giuliodori et al., 2018; McKoy et al., 2000; Pilichou et al., 2006; Rampazzo et al., 2002; Syrris et al., 2006), as well as cardiac and bone marrow mesenchymal stromal cells (Scalco et al., 2021; Sommariva et al., 2016; Stadiotti et al., 2021), also affects cSNs, at least in the AC forms linked to DSG2 mutations. An investigation of the effects of *Dsg2* mutation on SN physiology is beyond the scope of the present study. However, morphological SN alterations, clinical evidence of increased plasma levels of neuropeptide Y in patients (Stadiotti et al., 2021), higher expression of neuropeptide Y receptors in patient-derived AC stromal cells (Stadiotti et al., 2021) and a well-established association between stress, disease progression and life-threatening arrhythmias (Agrimi et al., 2020; Chelko et al., 2016; Corrado et al., 2003; Corrado et al., 2006) all suggest that SNs may be dysfunctional in AC patients. In the present study, we mainly focused on cSNs, although we expect that AC-linked mutations may affect SNs both within and outside the heart, therefore supporting the notion that AC is not limited to 'desmosome-carrying' cells, but develops with the contribution of several 'desmosomal protein-carrying' cell types and systems. In addition, the present study provides a basis for investigating the effects of other mutant desmosomal proteins in cardiac and extracardiac SNs, which may contribute to the variable manifestations of the different AC forms.

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

## Additional information

### Data availability statement

All raw data associated with the present study are provided within the published article.

### Competing interests

The authors declare that they have no competing interests.

### Author contributions

I.P.V. and A.S. performed *ex vivo* and *in vitro* experiments. I.P.V. analysed the 3-D reconstruction of the neuronal network. I.P.V. and A.S. interpreted results. I.P.V. and S.P.C. contributed to manuscript preparation. A.S. analysed data. M.R. contributed to biochemical and *in vitro* analyses. A.D.B. performed immunofluorescence analysis. C.O. performed whole-mount immunofluorescence in tissue clarified heart blocks and relative imaging analysis. S.R. provided human ganglia samples. S.P.C. provided A.C. mice. S.P.C., D.C., L.S. and C.B. critically discussed data. C.B. provided human samples. M.M. and T.Z. designed and supervised the study. M.M. and T.Z. interpreted and discussed the results. M.M. and T.Z. wrote the manuscript.

### Funding

This work was supported by Italian Ministerial grants PRIN 2022F3NENH to MM; PRIN 20229PX74A to TZ and PRIN 202249XEA5 to CB. IPV is supported by the European Union's Horizon 2020 research and innovation programme under the Marie Skłodowska-Curie grant agreement No 101 034 319 and from the European Union – NextGenerationEU.

### Acknowledgements

Open access publishing facilitated by Universita degli Studi di Padova, as part of the Wiley - CRUI-CARE agreement.

### Keywords

arrhythmogenic cardiomyopathy, arrhythmias, cardiac sympathetic neurons, desmoglein-2, sudden cardiac death

## Supporting information

Additional supporting information can be found online in the Supporting Information section at the end of the HTML view of the article. Supporting information files available:

**Peer Review History**

