## [Peer Review History · The Journal of Physiology]

Cardiac sympathetic neurons are additional cells affected in genetically determined Arrhythmogenic Cardiomyopathy

Induja Perumal Vanaja, Arianna Scalco, Marco Ronfini, Anna Di Bona, Camilla Olianti, Stefania Rizzo, Stephen P. Chelko, Domenico Corrado, Leonardo Sacconi, Cristina Basso, Marco Mongillo, and Tania Zaglia
DOI: 10.1113/JP286845

Corresponding author(s): Tania Zaglia (tania.zaglia@unipd.it)

Review Timeline:

Submission Date:	30-Apr-2024
Editorial Decision:	30-May-2024
Revision Received:	19-Jun-2024
Accepted:	05-Jul-2024

Senior Editor: Harold Schultz

Reviewing Editor: David Paterson

Transaction Report:

Dear Professor Zaglia,

Re: JP-RP-2024-286845 "Cardiac sympathetic neurons are additional cells affected in genetically determined Arrhythmogenic Cardiomyopathy" by Induja Perumal Vanaja, Arianna Scalco, Anna Di Bona, Camilla Olianti, Stefania Rizzo, Stephen P. Chelko, Domenico Corrado, Leonardo Sacconi, Cristina Basso, Marco Mongillo, and Tania Zaglia

Thank you for submitting your manuscript to The Journal of Physiology. It has been assessed by a Reviewing Editor and by 2 expert referees and we are pleased to tell you that it is acceptable for publication following satisfactory revision.

REVISION CHECKLIST:

Please upload two versions of your manuscript text: one with all relevant changes highlighted and one clean version with no changes tracked. The manuscript file should include all tables and figure legends, but each figure/graph should be uploaded as separate, high-resolution files. The journal is now integrated with Wiley's Image Checking service. For further details, see: <https://www.wiley.com/en-us/network/publishing/research-publishing/trending-stories/upholding-image-integrity-wileys-image-screening-service>.

- 'Potential Cover Art' for consideration as the issue's cover image
- Appropriate Supporting Information (video, audio or data set: see https://jp.msubmit.net/cgi-bin/main.plex?form_type=display_requirements#supp)

We look forward to receiving your revised submission.

Yours sincerely,

Harold Schultz
Senior Editor
The Journal of Physiology

REQUIRED ITEMS

- Author photo and profile. First or joint first authors are asked to provide a short biography (no more than 100 words for one author or 150 words in total for joint first authors) and a portrait photograph. These should be uploaded and clearly labelled together in a Word document with the revised version of the manuscript. See Information for Authors for further details.

- You must start the Methods section with a paragraph headed Ethical Approval. A detailed explanation of journal policy and regulations on animal experimentation is given in Principles and standards for reporting animal experiments in The Journal of Physiology and Experimental Physiology by David Grundy *J Physiol*, 593: 2547-2549. doi:10.1113/JP270818). A checklist outlining these requirements and detailing the information that must be provided in the paper can be found at: <https://physoc.onlinelibrary.wiley.com/hub/animal-experiments>. Authors should confirm in their Methods section that their experiments were carried out according to the guidelines laid down by their institution's animal welfare committee, and conform to the principles and regulations as described in the Editorial by Grundy (2015), including an ethics approval reference number. The Methods section must contain a statement about access to food, water and housing, details of the anaesthetic regime: anaesthetic used, dose and route of administration, and method of killing the experimental animals.

If experiments were conducted on humans, confirmation that informed consent was obtained, preferably in writing, that the studies conformed to the standards set by the latest revision of the Declaration of Helsinki and that the procedures were approved by a properly constituted ethics committee, which should be named, must be included in the article file. If the research study was registered (clause 35 of the Declaration of Helsinki), the registration database should be indicated, otherwise the lack of registration should be noted as an exception (e.g. The study conformed to the standards set by the Declaration of Helsinki, except for registration in a database). For further information see: <https://physoc.onlinelibrary.wiley.com/hub/human-experiments>.

- Your manuscript must include a complete Additional Information section, including competing interests; funding; author contributions and acknowledgements.

- Please upload separate high-quality figure files via the submission form.

- Please ensure that any tables are editable and in Word format, and wherever possible, embedded in the article file itself.

- Please ensure that the Article File you upload is a Word file.

- Your paper contains Supporting Information of a type that we no longer publish, including supplementary tables and figures. Any information essential to an understanding of the paper must be included as part of the main manuscript and figures. The only Supporting Information that we publish are video and audio, 3D structures, program codes and large data

files. Your revised paper will be returned to you if it does not adhere to our Supporting Information Guidelines.

- Papers must comply with the Statistics Policy: https://jp.msubmit.net/cgi-bin/main.plex?form_type=display_requirements#statistics.

In summary:

- If n {less than or equal to} 30, all data points must be plotted in the figure in a way that reveals their range and distribution. A bar graph with data points overlaid, a box and whisker plot or a violin plot (preferably with data points included) are acceptable formats.

- If $n > 30$, then the entire raw dataset must be made available either as supporting information, or hosted on a not-for-profit repository, e.g. FigShare, with access details provided in the manuscript.

- 'n' clearly defined (e.g. x cells from y slices in z animals) in the Methods. Authors should be mindful of pseudoreplication.

- All relevant 'n' values must be clearly stated in the main text, figures and tables.

- The most appropriate summary statistic (e.g. mean or median and standard deviation) must be used. Standard Error of the Mean (SEM) alone is not permitted.

- Exact p values must be stated. Authors must not use 'greater than' or 'less than'. Exact p values must be stated to three significant figures even when 'no statistical significance' is claimed.

- A Data Availability Statement is required for all papers reporting original data. This must be in the Additional Information section of the manuscript itself. It must have the paragraph heading 'Data Availability Statement'. All data supporting the results in the paper must be either: in the paper itself; uploaded as Supporting Information for Online Publication; or archived in an appropriate public repository. The statement needs to describe the availability or the absence of shared data. Authors must include in their statement: a link to the repository they have used, or a statement that it is available as Supporting Information; reference the data in the appropriate sections(s) of their manuscript; and cite the data they have shared in the References section. Whenever possible, the scripts and other artefacts used to generate the analyses presented in the paper should also be publicly archived. If sharing data compromises ethical standards or legal requirements then authors are not expected to share it, but must note this in their statement. For more information, see our Statistics Policy.

- Please include an Abstract Figure file, as well as the Figure Legend text within the main article file. The Abstract Figure is a piece of artwork designed to give readers an immediate understanding of the research and should summarise the main conclusions. If possible, the image should be easily 'readable' from left to right or top to bottom. It should show the physiological relevance of the manuscript so readers can assess the importance and content of its findings. Abstract Figures should not merely recapitulate other figures in the manuscript. Please try to keep the diagram as simple as possible and without superfluous information that may distract from the main conclusion(s). Abstract Figures must be provided by authors no later than the revised manuscript stage and should be uploaded as a separate file during online submission labelled as File Type 'Abstract Figure'. Please also ensure that you include the figure legend in the main article file. All Abstract Figures should be created using BioRender. Authors should use The Journal's premium BioRender account to export high-resolution images. Details on how to use and access the premium account are included as part of this email.

EDITOR COMMENTS

Reviewing Editor:

This is a well presented manuscript that highlights several novel findings that need to be clearly articulated given reviewer 1's assessment.

Authors should reference the recent JP white paper they are co-authors on along with Herring et al Nat Review Cardiol 2019 in the opening paragraph of the introduction to provide general context with these in-depth reviews.

Please also see 'Required Items' above.

Senior Editor:

Comments for Authors to ensure the paper complies with the Statistics Policy (Required):

Sample size must be stated (x samples in x animals) in the figure legends. Actual p values must be shown in graphs, (including ns). Symbols are not acceptable.

Thank you for submission of your research article to the Journal of Physiology for consideration. The article has been reviewed by experts in the field and found to be potentially acceptable for publication pending adequate revision to address all of the concerns raised. Please address all comments from the external referees and reviewing editor as well as addressing the list of additional concerns below, and the requirements or publication in the journal including the statistical requirements as stated in the summary letter.

Editorial concerns

1. The supplementary methods and figures/tables need to be incorporated into the manuscript and numbered accordingly.

2. The Methods section must begin with a subsection titled Ethical Approval. Missing information: Human tissue must indicate that samples were obtained with prior informed consent from the patients, and the study conformed to the latest standards set by the Declaration of Helsinki. Please indicate basic demographics of the sample population, age, sex and health status. Please review: <https://physoc.onlinelibrary.wiley.com/hub/human-experiments>

Missing information: Animals ethics must indicate that investigators understand the ethical principles under which the journal operates and that their work complies with the animal ethics checklist as outlined by the journal. This section must describe how the tissue was harvested from the animals including the method of anaesthesia or killing must be stated. Animals must be killed using methods approved for that species, stage of development and size. Describe the euthanasia protocol in detail (anaesthetic(s) used, dose, route), and indices used to confirm death. Please review: <https://physoc.onlinelibrary.wiley.com/hub/animal-experiments>

3. The manuscript must follow the statistics policy: Actual p values must be shown in graphs, (including ns). Symbols are not acceptable.

4. Please indicate the RRIDs of the antibodies in Tables 1 and 2 if known.

5. We advise that the immunoblot bands illustrated in the figures not be so closely cropped. In some cases the band itself is cropped. It is best practice to retain some space above and below the band of interest from the original image of gels or blots to show whether any nearby bands are present. It is not appropriate to crop the panel very close to the band itself.

6. Figure 5 could benefit with a scale bar.

7. We advise that the abstract be one continuous paragraph without subsections.

8. An abstract figure and legend is required. (please follow the guidelines outlined for the figure: do not simply show a figure from the paper)

9. The manuscript must include a data availability statement: https://jp.msubmit.net/cgi-bin/main.plex?form_type=display_requirements#addinfo

REFeree COMMENTS

Referee #1:

The investigators stress, as part of the novelty of their study, that AC has been approached as a "disease of cardiomyocytes." Indeed, a number of genetic manipulations have demonstrated the importance of desmosomal proteins to the function of cardiac myocytes, and many of those studies have shown that there are myocyte-intrinsic changes under conditions of cardiac desmosomal deficiency that are consistent with changes observed in the hearts of patients affected with ACM. But there are also studies that have explored the role of other cell types. The investigators note some; others (in particular the work of the van Rooij lab (PMID: 34550725) also deserve mention. Overall, the notion that other cell types participate in the whole phenotype of ACM is not a novelty of the present study. The fact that there is innervation remodeling in this particular *dsg2* mouse, is the novelty of the paper.

Related to the point above: This reviewer understands that the authors wish to highlight the importance of their findings, but a bit of a broader context would seem appropriate. Indeed, a number of studies have shown changes in the electrophysiology and the intracellular calcium homeostasis of myocytes consequent to desmosomal deficiency. Some of those changes have been studied directly in the context of adrenergic regulation and/or exercise. Those studies are not in contradiction, nor do they obscure, the findings of this study. Yet, acknowledging them would help to provide context for the reader as to the possible impact that the findings that are presented in this study may have, within the framework of what we know about arrhythmogenesis in ACM.

The anatomical descriptions presented in this study are valuable, novel and thought-provoking. Yet, the authors do not present any functional studies indicating that the anatomical remodeling observed can actually affect function in a manner conducive to arrhythmia risk. Without such an analysis, the study has limited impact in terms of our understanding of the cellular mechanisms of arrhythmogenesis in AC-afflicted hearts.

There is solid evidence indicating that at least some of the clinical manifestations (and likely, the cellular/molecular changes) that result from desmosomal deficiencies are not universal, but vary depending on the affected gene. The authors should stress that their results should only be interpreted within the context of the few cases of DSG2 deficiency, and not in the context of the overall population currently covered under the umbrella term of arrhythmogenic cardiomyopathy.

Other comments:

Western blots in Figures 1 and 2: please specify the differences between lanes (1, 2, 3 etc; are these different hearts?). Also, please show the entire lane. The small windows prevent the reader from evaluating the specificity (or lack thereof) of the antibody.

Figures 1B-C: There appears to be substantial DSG2 staining coming from the nuclei. Is that a common observation in cells that express DSG2? Could that be an artifact due to non-specific binding of the antibody? Evidence of the selectivity of the antibody should be provided.

Figure 5: How many hearts were included? Where are the chambers in this projection? I would have expected blank space where the chambers are. I really do not see differences that could not be explained by variability between samples. The authors may consider a better explanation for this figure, a quantitative analysis for it, or its removal from the dataset.

Referee #2:

The Zaglia group present an interesting study regarding an arrhythmogenic cardiomyopathy mutation in DSG2, which as well as effecting cardiomyocyte coupling particularly in the left ventricle, also leads to heterogeneous sympathetic hyperinnervation. Given, that this condition is associated with ventricular arrhythmia during sympathetic stress, this is an important finding and significant conceptual advancement in the field.

The strength of the study is the detailed confocal immunofluorescence and multiphoton imaging of clarified heart blocks in control and mutant mice, along with the recapitulation of the phenotype in neonatal isolated sympathetic neurons with the mutant prior to cardiac innervation, and induction of the phenotype through downregulation of DSG2 in normal PC12 cells. In addition, the expression of DSG2 was confirmed in human stellate ganglia tissue.

The weakness of the study is the lack of mechanism regarding how a protein previously considered to solely be part of the cardiomyocyte desmosome, influences sympathetic neuronal innervation.

Other points to address:

1. It is interesting that the DSG2 mutation led to decreased neurite length and branching in culture, whereas the tissue clearing experiments suggest a heterogeneous increase in TH fibre number, length and density. This is not made clear in the abstract and start of the discussion where only "alterations" are repeatedly referred to with regards to the findings in culture. This should be described more specifically. Do the authors have an explanation for this seemingly contradictory result between the findings in culture and in the intact heart?

2. How many hearts underwent tissue clearing and thin sectioning in each group? How many cells were analysed in culture from how many animals? The n numbers throughout the manuscript in terms of whether they refer to cells/slices vs animals/ (ie technical vs biological repeats) is unclear.

3. The abstract suggests that DSG2 downregulation induced a similar phenotype in sympathetic neurons. This is misleading as the sh-RNA experiments were carried out in differentiated PC12 cells (not isolated "sympathetic neurons" from the SCG and stellate ganglia).

4. A scrambled sh-RNA should be used as a control rather than an empty vector for the knockdown experiments.

5. It is worth noting that sympathetic hyperinnervation has also recently been identified in a mouse model of CPVT (secondary to RyR2 mutation R2474S) by O'Reilly et al and this should be cited (<https://doi.org/10.1093/cvr/cvae088.098>)

6. The method of animal euthanasia is unclear and need to be stated according to the journal standards.

END OF COMMENTS

1222·2022
800
A N N I

UNIVERSITÀ
DEGLI STUDI
DI PADOVA

DIPARTIMENTO DI SCIENZE BIOMEDICHE – DSB
DEPARTMENT OF BIOMEDICAL SCIENCES

Via Ugo Bassi 58/B
35131 Padova – Italy
dipartimento.biomed@pec.unipd.it
CF 80006480281
P.IVA 00742430283
www.biomed.unipd.it

Padova, 19 June 2024

Dear Editor,

we here submit our revised version of the manuscript entitled " **Cardiac sympathetic neurons are additional cells affected in genetically determined Arrhythmogenic Cardiomyopathy** ", for possible publication in *the Journal of Physiology*.

All reviewers' concerns have been addressed, as detailed in the 'point-by-point' response.

We declare that all co-authors have read the manuscript and agree with its content; that none of the authors has any conflict of interest or financial interest; that the submission is novel, and it is not under review in any other journal.

We hope that you are willing to receive our manuscript for possible publication in your journal and look forward to hearing from you.

Sincerely yours,

Tania Zaglia, PhD

Department of Biomedical Sciences, University of Padua
Via Ugo Bassi 58/B, 35131 Padova, Italy
e-mail: tania.zaglia@unipd.it
phone number: +390497923294
fax number: +390497923250

POINT-BY-POINT RESPONSE

JP-RP-2024-286845 "Cardiac sympathetic neurons are additional cells affected in genetically determined Arrhythmogenic Cardiomyopathy"

EDITOR COMMENTS

Reviewing Editor: This is a well-presented manuscript that highlights several novel findings that need to be clearly articulated given reviewer 1's assessment. Authors should reference the recent JP white paper they are co-authors on along with Herring et al Nat Review Cardiol 2019 in the opening paragraph of the introduction to provide general context with these in-depth reviews. Please also see 'Required Items' above.

Senior Editor: Comments for Authors to ensure the paper complies with the Statistics Policy (Required): Sample size must be stated (x samples in x animals) in the figure legends. Actual p values must be shown in graphs, (including ns). Symbols are not acceptable.

Au: We thank the Reviewing and Senior Editors for their appreciation of the manuscript message. We also thank the reviewers for their comments which we addressed, as detailed below, to increase the quality of this manuscript. In addition, Marco Ronfini has been added among the authors as he contributed to the manuscript revision.

EDITORIAL CONCERNS

1) The supplementary methods and figures/tables need to be incorporated into the manuscript and numbered accordingly.

Au: Supporting data has been included in the main text.

2) The Methods section must begin with a subsection titled Ethical Approval. Missing information: Human tissue must indicate that samples were obtained with prior informed consent from the patients, and the study conformed to the latest standards set by the Declaration of Helsinki. Please indicate basic demographics of the sample population, age, sex and health status. Please review: <https://physoc.onlinelibrary.wiley.com/hub/human-experiments>.

Au: Amended accordingly.

"Human tissue sample processing and immunofluorescence. We analyzed postmortem stellate ganglia samples from two male patients (age: 50±3 years) died for extra-cardiac causes (accidents), who did not have prior history of heart disease. Samples were archived in the historical collection of the Institute of Pathological Anatomy of the University of Padova and were acquired during routine postmortem investigations. Samples were anonymized to the investigators and were used in accordance with the directives of the national committee of Bioethics and "Raccomandazione (2006) della Commissione dei Ministri degli Stati Membri sull'utilizzo di campioni biologici di origine umana per scopi di ricerca". Samples were analyzed by confocal immunofluorescence using the protocol previously described by Zaglia et al., (2016)."

Missing information: Animals ethics must indicate that investigators understand the ethical principles under which the journal operates and that their work complies with the animal ethics checklist as outlined by the journal. This section must describe how the tissue was harvested from the animals including the method of anesthesia or killing must be stated. Animals must be killed using methods approved for that species, stage of development and size. Describe the euthanasia protocol in detail (anesthetic(s) used, dose, route), and indices used to confirm death. Please review:

<https://physoc.onlinelibrary.wiley.com/hub/animal-experiments>.

Au: Amended accordingly.

"Ethical Approval. All of the investigators involved in the present study understand the ethical principles under which the journal operates, and the work conducted complies with the animal ethics checklist of The Journal of Physiology (Grundy, 2015)."

"Mice were killed by cervical dislocation (in accordance with Annex IV of European Directive 2010/63/EU)."

3) The manuscript must follow the statistics policy: Actual p values must be shown in graphs, (including ns). Symbols are not acceptable.

Au: Amended accordingly.

4) Please indicate the RRIDs of the antibodies in Tables 1 and 2 if known.

Au: Amended accordingly.

5) We advise that the immunoblot bands illustrated in the figures not be so closely cropped. In some cases, the band itself is cropped. It is best practice to retain some space above and below the band of interest from the original image of gels or blots to show whether any nearby bands are present. It is not appropriate to crop the panel very close to the band itself.

Au: Amended accordingly.

6) Figure 5 could benefit with a scale bar.

Au: Amended accordingly.

7) We advise that the abstract be one continuous paragraph without subsections.

Au: Amended accordingly.

8) An abstract figure and legend are required. (please follow the guidelines outlined for the figure: do not simply show a figure from the paper)

Au: Amended accordingly.

9) The manuscript must include a data availability statement: https://jp.msubmit.net/cgi-bin/main.plex?form_type=display_requirements#addinfo

Au: Amended accordingly.

REFeree COMMENTS

Referee #1: The investigators stress, as part of the novelty of their study, that AC has been approached as a "disease of cardiomyocytes." Indeed, a number of genetic manipulations have demonstrated the importance of desmosomal proteins to the function of cardiac myocytes, and many of those studies have shown that there are myocyte-intrinsic changes under conditions of cardiac desmosomal deficiency that are consistent with changes observed in the hearts of patients affected with ACM. But there are also studies that have explored the role of other cell types. The investigators note some; others (in particular the work of the van Rooij lab (PMID: 34550725) also deserve mention. Overall, the notion that other cell types participate in the whole phenotype of ACM is not a novelty of the present study. The fact that there is innervation remodeling in this particular *dsg2* mouse, is the novelty of the paper.

1) Related to the point above: This reviewer understands that the authors wish to highlight the importance of their findings, but a bit of a broader context would seem appropriate. Indeed, a number of studies have shown changes in the electrophysiology and the intracellular calcium homeostasis of myocytes consequent to desmosomal deficiency. Some of those changes have been studied directly in the context of adrenergic regulation and/or exercise. Those studies are not in contradiction, nor do they obscure, the findings of this study. Yet, acknowledging them would help to provide context for the reader as to the possible impact that the findings that are presented in this study may have, within the framework of what we know about arrhythmogenesis in ACM.

Au: We understand the reviewer's comment and modified the introduction and the discussion according to his/her suggestions (see below).

Introduction: "The typical myocardial remodeling and the association of the disease with desmosomal gene mutations have steered AC research towards cardiomyocytes, the quintessential desmosome-carrying cells in the heart. However, several non-cardiomyocyte cell types have recently been implicated in the typical AC remodeling, including: resident cardiac stem cells (Lombardi et al., 2011); progenitor

cells from the second heart field (Lombardi et al., 2009); epicardium-derived progenitors (Matthes et al., 2011; Kohela et al., 2021) and mesenchymal stromal cells (Sommariva et al., 2016; Stadiotti et al., 2021). Notably, current literature and our data indicate that almost all cardiac and extra-cardiac cell types express desmosomal proteins and harbor therefore the AC-linked mutant variants (Sommariva et al., 2016; Scalco et al., 2021; Stadiotti et al., 2021). On these bases, we inquired whether desmosome-linked AC may affect the entire myocardial cell network”

Discussion: “AC is indeed characterized by life-threatening ventricular arrhythmias, which several studies in animal and human models have attributed to the effects of AC-linked mutations on cardiomyocyte electrophysiology. In detail, it has been demonstrated that destabilization of desmosomes leads to electrical remodeling, caused by alterations in gap junction organization and connexin expression (Oxford et al., 2007; Gehmlich et al., 2011; Rizzo et al., 2012; Chevalier et al., 2021; Stevens et al., Cells, 2022; Reisqs et al., 2023). Perturbation of the desmosomal structure can also lead to the loss of voltage-gated sodium channels (Nav1.5), which often occurs in hearts with minor or undetectable structural lesions (Rizzo et al., 2012; Cerrone & Delmar., 2014; Zaklyazminskaya & Dzemeshkevich., 2016; Reisqs et al., 2023). In addition, mutations in desmosomal proteins affect cardiomyocyte expression of genes controlling Ca²⁺ dynamics, causing arrhythmogenic alteration in Ca²⁺ handling (Cerrone et al., 2017; Tiso et al., 2001; van der Zwaag et al., 2012; Reisqs et al., 2023 Vallverdù-Prats et al.,2023). Such changes have also been described in the context of physical (i.e. exercise) or emotional (i.e. psychosocial stress) stresses, both conditions triggering life-threatening arrhythmias and independently accelerating AC progression (Wichter et al., 2000; Corrado et al., 2003; Saberniak et al., 2014; Asimaki et al., 2015; Corrado & Zorzi., 2015; Chelko et al., 2016; Martherus et al., 2016; Corrado et al., 2017; Corrado et al., 2019; Agrimi et al., 2020). Notably, exercise and emotions are well-known activators of the Sympathetic Nervous System, whose activation may favor the onset of reentry mechanisms around the fibro-fatty myocardial areas (Ripplinger et al 2016; Herring et al 2019; Habecker et al., 2024; Zaglia & Mongillo., 2017).”

2) The anatomical descriptions presented in this study are valuable, novel and thought-provoking. Yet, the authors do not present any functional studies indicating that the anatomical remodeling observed can actually affect function in a manner conducive to arrhythmia risk. Without such an analysis, the study has limited impact in terms of our understanding of the cellular mechanisms of arrhythmogenesis in AC-afflicted hearts.

Au: We appreciate that the reviewer finds the anatomical description of the alterations of cardiac sympathetic innervation novel and thought-provoking. We are aware that our study focuses on morphological description, which is however a required observation to set the ground for further studies aimed at assessing how desmosomal mutations may affect sympathetic neuron function. The link between alterations in cardiac sympathetic neuron distribution and increased arrhythmogenesis is well acknowledged, and now better discussed in the text (Ripplinger et al., 2016; Herring et al 2019; Habecker et al., 2024; Zaglia & Mongillo., 2017). As such, our findings, that cardiac innervation in *Dsg2* mutant hearts is dramatically altered, strongly support a possible involvement of neurons in AC.

3) There is solid evidence indicating that at least some of the clinical manifestations (and likely, the cellular/molecular changes) that result from desmosomal deficiencies are not universal but vary depending on the affected gene. The authors should stress that their results should only be interpreted within the context of the few cases of DSG2 deficiency, and not in the context of the overall population currently covered under the umbrella term of arrhythmogenic cardiomyopathy.

Au: We totally agree with the reviewer comment, and we pointed out this concept in the new version of the discussion.

“... also hits cSNs, at least in the AC forms linked to DSG2 mutations”.

“...In addition, this study poses the bases for studying the effects of other mutant desmosomal proteins in cardiac and extracardiac SNs, which may contribute to the variable manifestations of the different AC forms”.

Other comments:

4) Western blots in Figures 1 and 2: please specify the differences between lanes (1, 2, 3 etc; are these different hearts?). Also, please show the entire lane. The small windows prevent the reader from evaluating the specificity (or lack thereof) of the antibody.

Au: The numbers 1, 2, etc, in western blotting images indicate different samples. We specified this in the corresponding Figure Legend. In addition, images of western blotting in the Figures have been modified following the reviewer's suggestions.

5) Figures 1B-C: There appears to be substantial DSG2 staining coming from the nuclei. Is that a common observation in cells that express DSG2? Could that be an artifact due to non-specific binding of the antibody? Evidence of the selectivity of the antibody should be provided.

Au: To exclude artifacts of the primary or secondary antibodies, we added images of control experiments in which we used the secondary antibody alone and we added in Figure 3C a high magnification of immunofluorescence staining in murine stellate ganglia from normal mice, showing the co-existence of cells with DSG2 nuclear expression with DGS2-negative cells without nuclear signaling. The biological significance of such finding is beyond the scope of the current work and will be investigated in following research.

6) Figure 5: How many hearts were included? Where are the chambers in this projection? I would have expected blank space where the chambers are. I really do not see differences that could not be explained by variability between samples. The authors may consider a better explanation for this figure, a quantitative analysis for it, or its removal from the dataset.

Au: Figure 9 (old Figure 5) shows a representative image of one ventricular heart section from a *Dsg2^{WT/WT}* mouse (panel A) and one representative section from a *Dsg2^{mut/mut}* mouse (panel B). A total of 5 hearts/group have been analyzed and for each heart we analyzed four non-consecutive sections. This Picture represents in pseudocolors the density of TH-positive fibers in the different regions of the heart and aims to show the altered topology and heterogeneous distribution of SNs in AC hearts. Quantification in all the heart dataset is shown in Figure 8. To increase clarity, we added a paragraph in the Method section describing the procedure used to correlate SN density to colors (see below). In addition, ventricular chambers are now indicated as a white space delimited by a red line.

“Colorimetric map generation. Colorimetric maps were used to visualize SN distribution within sections from the mid portion of the ventricles of 6 months old *Dsg2^{WT/WT}* and *Dsg2^{mut/mut}* mice. Five hearts/group were evaluated and for each heart, we analyzed four non-consecutive sections. Sections were stained with an antibody to Tyrosine Hydroxylase and images acquired at the fluorescent microscope (Leica DM6B, Leica GmbH) were composed with Fiji (Schindelin et al., 2012). Average fluorescence intensity was calculated in image bins of 150 by 150-pixel units, to obtain a spatially resolved semi-quantitative image of neuronal distribution and were represented in pseudo-colour scale ranging from blue (i.e. low-density) to red (i.e. high-density).”

Referee #2: The Zaglia group presents an interesting study regarding an arrhythmogenic cardiomyopathy mutation in DSG2, which as well as effecting cardiomyocyte coupling particularly in the left ventricle, also leads to heterogeneous sympathetic hyperinnervation. Given, that this condition is associated with ventricular arrhythmia during sympathetic stress, this is an important finding and significant conceptual advancement in the field. The strength of the study is the detailed confocal immunofluorescence and multiphoton imaging of clarified heart blocks in control and mutant mice, along with the recapitulation of the phenotype in neonatal isolated sympathetic neurons with the mutant prior to cardiac innervation, and induction of the phenotype through downregulation of DSG2 in normal PC12 cells. In addition, the expression of DSG2 was confirmed in human stellate ganglia tissue.

Au: We thank the reviewer for acknowledging that our work provides important findings in the field of AC research. We also understand his/her comments, which have been addressed as detailed below.

1) The weakness of the study is the lack of mechanism regarding how a protein previously considered to solely be part of the cardiomyocyte desmosome, influences sympathetic neuronal innervation.

Au: We are aware that our study does not provide mechanistic insight to explain how DSG2 mutations affect the neuronal phenotype. However, our results demonstrate unambiguously that in a pathophysiological relevant disease model, sympathetic neurons are directly affected by the DSG2 mutation.

As in other cell types, DSG2 does not establish desmosomal connections in sympathetic neurons. Indeed, our IF analyses showed a diffuse cytosolic and nuclear protein localization. We believe that in 'non-desmosome carrying cells', desmosomal proteins have a key role in cell signaling as previously showed by us (Stadiotti et al., 2021; Scalco et al., 2022) and others (Lombardi et al., 2011). As assessed in the discussion, the role of DSG2 in sympathetic neuron biology is the object of a research on-going in the laboratory.

In addition, at further proof of the non-desmosome related role of DSG2, the protein's highest expression is in the colon [<https://www.ncbi.nlm.nih.gov/gene/1829>]. According to the NIH database, DSG2 is expressed ubiquitously throughout the body, and of the n=27 organs listed by the NIH, cardiac DSG2 is in the 7th position (i.e. 6 other organs have higher DSG2 expression compared to the heart).

Other points to address:

2) It is interesting that the DSG2 mutation led to decreased neurite length and branching in culture, whereas the tissue clearing experiments suggest a heterogeneous increase in TH fiber number, length and density. This is not made clear in the abstract and start of the discussion where only "alterations" are repeatedly referred to with regards to the findings in culture. This should be described more specifically. Do the authors have an explanation for this seemingly contradictory result between the findings in culture and in the intact heart?

Au: We thank the reviewer for the suggestion and amended the abstract accordingly.

We agree with the reviewer' observation that *in vitro* and *in vivo* results are apparently contradictory, and we added comments on this matter in the current version of the discussion (see below).

"This work by combining *ex vivo* analyses on murine and human stellate ganglia samples, multiphoton imaging of clarified heart blocks and *in vitro* assays in cultured neurons demonstrates that AC-linked *Dsg2* mutations lead to significant effects on cardiac autonomic innervation."

"Taken together experimental evidence that DSG2 has a primary role in SN biology, the different phenotype of SNs in the *in vitro* vs. the *in vivo* contexts, and the disease stage-dependency of sympathetic neuropathology of the AC hearts suggest that altered cardiac sympathetic innervation may result from the combination of cell-autonomous (i.e. primary effect of DSG2 mutation in SNs) and context-dependent factors (i.e. aberrant release of chemorepellent/chemoattractant factors from mutant AC myocardium) implicated in myocardial remodeling. This latter point is in line with increasing evidence whereby primary myocardial damage may secondarily affect cSNs (Habecker et al., 2016; Dokshokova et al., 2022)."

3) How many hearts underwent tissue clearing and thin sectioning in each group? How many cells were analysed in culture from how many animals? The n numbers throughout the manuscript in terms of whether they refer to cells/slices vs animals/ (ie technical vs biological repeats) is unclear.

Au: We understand the reviewer' concern and amended Figure Legends accordingly.

4) The abstract suggests that DSG2 downregulation induced a similar phenotype in sympathetic neurons. This is misleading as the sh-RNA experiments were carried out in differentiated PC12 cells (not isolated "sympathetic neurons" from the SCG and stellate ganglia).

Au: Amended accordingly.

"Consistently, virus-assisted DSG2 downregulation replicated, in PC12-derived SNs, the phenotypic alterations displayed by *Dsg2*^{mut/mut} primary neurons...."

5) A scrambled sh-RNA should be used as a control rather than an empty vector for the knockdown experiments.

Au: We agree with the reviewer. Indeed, in our experiments we used both controls, but we originally presented only data of Ad-Empty cells as we did not observe significant differences between scrambled sh-RNA vs. Ad-Empty infected sympathetic neurons. In the revised version of the manuscript, we added both controls.

6) It is worth noting that sympathetic hyperinnervation has also recently been identified in a mouse model of CPVT (secondary to RyR2 mutation R2474S) by O'Reilly et al and this should be cited (<https://doi.org/10.1093/cvr/cvae088.098>)

Au: Amended accordingly.

“Interestingly, recent evidence shows that sympathetic hyperinnervation occurs in a stress-related arrhythmic syndrome model, i.e. Catecholaminergic Polymorphic Ventricular Tachycardia, which only shares with AC, the stress-dependency of arrhythmias, and none of the structural remodeling features (<https://doi.org/10.1093/cvr/cvae088.098>).”

7) The method of animal euthanasia is unclear and need to be stated according to the journal standards.

Au: Amended accordingly.

Dear Dr Zaglia,

Re: JP-RP-2024-286845R1 "Cardiac sympathetic neurons are additional cells affected in genetically determined Arrhythmogenic Cardiomyopathy" by Induja Perumal Vanaja, Arianna Scalco, Marco Ronfini, Anna Di Bona, Camilla Olianti, Stefania Rizzo, Stephen P. Chelko, Domenico Corrado, Leonardo Sacconi, Cristina Basso, Marco Mongillo, and Tania Zaglia

We are pleased to tell you that your paper has been accepted for publication in The Journal of Physiology.

Authors should note that it is too late at this point to offer corrections prior to proofing. Major corrections at proof stage, such as changes to figures, will be referred to the Editors for approval before they can be incorporated. Only minor changes, such as to style and consistency, should be made at proof stage. Changes that need to be made after proof stage will usually require a formal correction notice.

If you would like to receive our 'Research Roundup', a monthly newsletter highlighting the cutting-edge research published in The Physiological Society's family of journals (The Journal of Physiology, Experimental Physiology and Physiological Reports), please click this link, fill in your name and email address and select 'Research Roundup': <https://www.physoc.org/journals-and-media/membernews/>.

Yours sincerely,

Harold Schultz
Senior Editor
The Journal of Physiology

P.S. - You can help your research get the attention it deserves! Check out Wiley's free Promotion Guide for best-practice recommendations for promoting your work at www.wileyauthors.com/eeo/guide. You can learn more about Wiley Editing Services which offers professional video, design, and writing services to create shareable video abstracts, infographics, conference posters, lay summaries, and research news stories for your research at www.wileyauthors.com/eeo/promotion.

IMPORTANT NOTICE ABOUT OPEN ACCESS: To assist authors whose funding agencies mandate public access to published research findings sooner than 12 months after publication, The Journal of Physiology allows authors to pay an Open Access (OA) fee to have their papers made freely available immediately on publication.

You can check if your funder or institution has a Wiley Open Access Account here: <https://authorservices.wiley.com/author-resources/Journal-Authors/licensing-and-open-access/open-access/author-compliance-tool.html>.

EDITOR COMMENTS

Reviewing Editor:

The authors have done a good job revising the manuscript.

Senior Editor:

The editors wish to thank the authors for these final adjustments to the manuscript. The article is now accepted for publication. Congratulations for an interesting and insightful study. Please consider the Journal of Physiology for your future studies.

REFEREE COMMENTS

Referee #1:

I have no further comments

Referee #2:

The authors have addressed my points and the manuscript is improved.